# Bioreactor-based mass production of human iPSC-derived macrophages enables immunotherapies against bacterial airway infections

Mania Ackermann[1,2], Henning Kempf [3,12], Miriam Hetzel[2,4], Christina Hesse[5,6], Anna Rafiei Hashtchin[1,2], Kerstin Brinkert[7], Juliane Wilhelmine Schott[2], Kathrin Haake[1,2], Mark Philipp Kühnel[6,8], Silke Glage[9], Constanca Figueiredo[10], Danny Jonigk[6,8], Katherina Sewald[5,6], Axel Schambach[2,11], Sabine Wronski[5,6], Thomas Moritz[2,4], Ulrich Martin [3,6], Robert Zweigerdt [3], Antje Munder [6,7] & Nico Lachmann [1,2]

The increasing number of severe infections with multi-drug-resistant pathogens worldwide highlights the need for alternative treatment options. Given the pivotal role of phagocytes and especially alveolar macrophages in pulmonary immunity, we introduce a new, cell-based treatment strategy to target bacterial airway infections. Here we show that the mass production of therapeutic phagocytes from induced pluripotent stem cells (iPSC) in industry-compatible, stirred-tank bioreactors is feasible. Bioreactor-derived iPSC-macrophages (iPSC-Mac) represent a highly pure population of $CD45^+CD11b^+CD14^+CD163^+$ cells, and share important phenotypic, functional and transcriptional hallmarks with professional phagocytes, however with a distinct transcriptome signature similar to primitive macrophages. Most importantly, bioreactor-derived iPSC-Mac rescue mice from *Pseudomonas aeruginosa*-mediated acute infections of the lower respiratory tract within 4-8 h post intra-pulmonary transplantation and reduce bacterial load. Generation of specific immune-cells from iPSC-sources in scalable stirred-tank bioreactors can extend the field of immunotherapy towards bacterial infections, and may allow for further innovative cell-based treatment strategies.

[1] JRG Translational Hematology of Congenital Diseases, Hannover Medical School, Hannover, Germany. [2] Institute of Experimental Hematology, REBIRTH Cluster of Excellence, Hannover Medical School, 30625 Hannover, Germany. [3] Leibniz Research Laboratories for Biotechnology and Artificial Organs (LEBAO), Department of Cardiothoracic, Transplantation and Vascular Surgery, REBIRTH Cluster of Excellence, Hannover Medical School, 30625 Hannover, Germany. [4] RG Reprogramming and Gene Therapy, Hannover Medical School, 30625 Hannover, Germany. [5] Fraunhofer Institute for Toxicology and Experimental Medicine (ITEM), REBIRTH Cluster-of Excellence, 30625 Hannover, Germany. [6] Biomedical Research in Endstage and Obstructive Lung Disease (BREATH), German Center for Lung Research, 30625 Hannover, Germany. [7] Clinical Research Group 'Cystic Fibrosis', Clinic for Pediatric Pneumology, Allergology and Neonatology, Hannover Medical School, 30625 Hannover, Germany. [8] Institute for Pathology, Hannover Medical School, 30625 Hannover, Germany. [9] Institute of Laboratory Animal Science and Central Animal Facility, Hannover Medical School, Hannover, Germany. [10] Institute for Transfusion Medicine, Hannover Medical School, 30625 Hannover, Germany. [11] Division of Hematology/Oncology, Boston Children's Hospital, Boston, MA 02215, USA. [12] Present address: Department of Stem Cell Biology, Novo Nordisk A/S, 2760 Maaloev, Denmark. These authors contributed equally: Mania Ackermann, Henning Kempf, Antje Munder, Nico Lachmann. Correspondence and requests for materials should be addressed to N.L. (email: lachmann.nico@mh-hannover.de)

nfectious diseases of the lower respiratory tract are exacerbated by the lack of sufficient treatment options, which is underlined by the 3.2 million reports of death in 2015 (World Health Organization; WHO). Although less frequently observed in wealthy countries, the increasing number of nosocomial infections constitutes a substantial public health problem worldwide[1]. Treatment of lower respiratory tract infections becomes particularly problematic, if the invasive microorganisms are resistant to standard or reserve antibiotic therapy as observed with increasing frequency for infections triggered by *Pseudomonas aeruginosa* (*P. aeruginosa*), one of the most common pathogens responsible for severe nosocomial infections[2]. In fact, the most recent global priority list of antibiotic-resistant bacteria to guide research, discovery, and development of new antibiotics from the WHO, states carbapenem-resistant *P. aeruginosa* as one of the three most critical pathogens for which new treatment options are urgently required[3].

One promising alternative to antibiotic therapy might be a cell-based immunotherapy approach applying phagocytes to enhance pulmonary immunity. As of yet, generating the therapeutically required amount of immune cells from peripheral blood or other sources remains challenging. In contrast to somatic cells, human-induced pluripotent stem cells (hiPSC), with their unlimited potential for proliferation and differentiation, may—in principle—enable this therapeutic scenario. In this line, hematopoietic differentiation of human iPSC has been proven feasible[4–6] and thus, has been proposed as a promising strategy for future cell-based treatment approaches. However, clinical translation of hiPSC-derived hematopoiesis remains hampered by (i) insufficient knowledge about in vivo functionality and (ii) lack of therapeutically required quantities of effector cells.

Considering phagocytes, and especially alveolar macrophages as critical regulators in the maintenance of lung homeostasis and pulmonary immunity[7–9], here we evaluate the therapeutic potential of iPSC-derived macrophages (iPSC-Mac) for the treatment of pulmonary infections caused by *P. aeruginosa*. We show that human iPSC can be efficiently differentiated into primitive macrophages in scalable suspension culture, and that the differentiation and harvesting process is transferable to industry-compatible stirred bioreactor systems. More importantly, we demonstrate that pulmonary transplantation of iPSC-Mac can prevent the onset of acute respiratory *P. aeruginosa* infection and rescue mice from established pulmonary infections and severe respiratory insufficiency.

## Results

**Mass production of human macrophages in stirred bioreactors.** Although the generation of different mature hematopoietic cell types from PSC has been proven successful using classical two-dimensional (2D) differentiation cultures[4,10–13], these systems do not allow for the generation of iPSC-derived cells in clinically relevant quantities. Thus, we developed a suspension-based (3D), continuous (4D) hematopoietic differentiation protocol, suitable for process upscaling in industry-compatible stirred tank bioreactors[14,15]. Using a well-characterized hiPSC line (hCD34iPSC16)[16], we induced the formation of myeloid cell forming complexes (MCFCs) from embryoid bodies (EBs) in small-scale suspension culture on an orbital shaker (Supplementary Fig. 1a and 1b). After 10–15 days, MCFCs continuously produced iPSC-Mac that could be harvested weekly for up to 3 months. Generated iPSC-Mac exhibited a clear surface marker profile of CD45+CD11b+CD14+CD163+CD34−TRA1-60−, although freshly collected cells comprised a minor population of CD45+/CD11b+/CD14−/CD163− immature myeloid cells (Supplementary Fig. 1c and d). Following terminal differentiation for 7

more days, iPSC-Mac represented a homogenous population of CD45+CD11b+CD14+CD163+CD34−TRA1-60− cells with classical macrophage-like morphology and efficiently phagocytosed fluorescently labeled *E. coli* particles (Supplementary Fig. 1e-g).

We then translated the suspension-based differentiation into stirred tank bioreactors using an industry-compatible system (DASbox Mini Bioreactor System)[17] previously applied for the efficient cultivation of human iPSC and their differentiation into cardiomyocytes[18] (Fig. 1a, b and Supplementary Fig. 2a). From day 10 onwards, weekly harvest of iPSC-Mac from the bioreactors showed an increase in cell yield over time, reaching a stable production of ~1–3 × 10$^7$ iPSC-Mac per week as early as in week 3, which was maintained for more than 5 weeks in two independent bioreactor runs (Fig. 1c, Supplementary Movie 1). Efficient generation of iPSC-Mac in both bioreactor experiments was reflected by the weekly increase in biomass, particularly during the first days after full medium refreshment. Dissolved oxygen (DO) and pH monitoring revealed expected zigzag-like patterns typical for repeated batch cultures. Notably, all process parameters showed maintenance of repetitive patterns after reaching the steady state of macrophage production around d15–20, confirming the overall stability of the process (Fig. 1d and Supplementary Fig. 2b). This finding was further supported by stable values for glucose, lactate, lactate dehydrogenase, and osmolality determined weekly parallel to macrophage harvests (Supplementary Fig. 2c). Similarly, secretion of cytokines/chemokines associated with the activation of macrophages, such as IL2, IL6, IL8, MCP1, TNF, and IFNα2, was detected from the first harvest (week 2) onwards (Fig. 1e) and corresponded with the appearance of CD45+ iPSC-Mac. MCFCs cultivated in the bioreactor sustained their morphology during the entire process and continuously generated iPSC-Mac of typical morphology and a CD45+CD14+ surface marker profile (Fig. 1f).

**Detailed characterization of bioreactor-derived iPSC-Mac.** Detailed characterization of unsorted, bioreactor-derived, and freshly harvested iPSC-Mac revealed a highly pure CD45+CD11b+CD163+CD14+CD34−TRA1-60− phenotype, and a typical morphology after adherence to tissue culture plates (Fig. 2a, b). Comparison of iPSC-Mac to non-differentiated hiPSC and peripheral blood mononuclear cells (PBMC)-derived macrophages (PBMC-Mac) by unsupervised, hierarchical heatmap clustering of whole transcriptomes revealed close proximity of iPSC-Mac and PBMC-Mac when compared to iPSC (Fig. 2c). Analysis of genes associated with pluripotency and activation of innate immune response, confirmed efficient differentiation of iPSC into macrophage-like cells (Fig. 2d). Importantly, genes associated with macrophage function such as the toll-like receptors (TLR) 1 and 4, CD14, or components of the NF-κB signaling pathway (gene ontology activation of innate immune response (GO: 0002218)) were significantly upregulated in iPSC-Mac and PBMC-Mac versus iPSC, whereas the opposite could be observed when analyzing genes associated with pluripotency (Fig. 2d). As expected, upregulated genes in iPSC-Mac versus pluripotent hiPSC were assigned to CD14+ monocytes, CD33+ myeloid cells and whole blood, including key genes, such as *CYBB*, *CSFR1*, *CCR1*, or the formyl peptide receptor 1 (*FPR1*).

Besides these global similarities between iPSC-Mac and PBMC-Mac, direct comparison between the two cell types revealed several differentially expressed genes (DEG) (Fig. 2e). Here, PBMC-Mac showed higher expression of HLA II molecules (*HLA-DQs*, *HLA-DRs*), as well as genes associated with CD56+ natural killer (NK) cells, such as granzyme A, K, and H. On the other hand, iPSC-Mac showed significant higher levels of extracellular matrix proteins, such as *CYR61*, *COL3A1*, *KRT 8/18/19* as well as several members

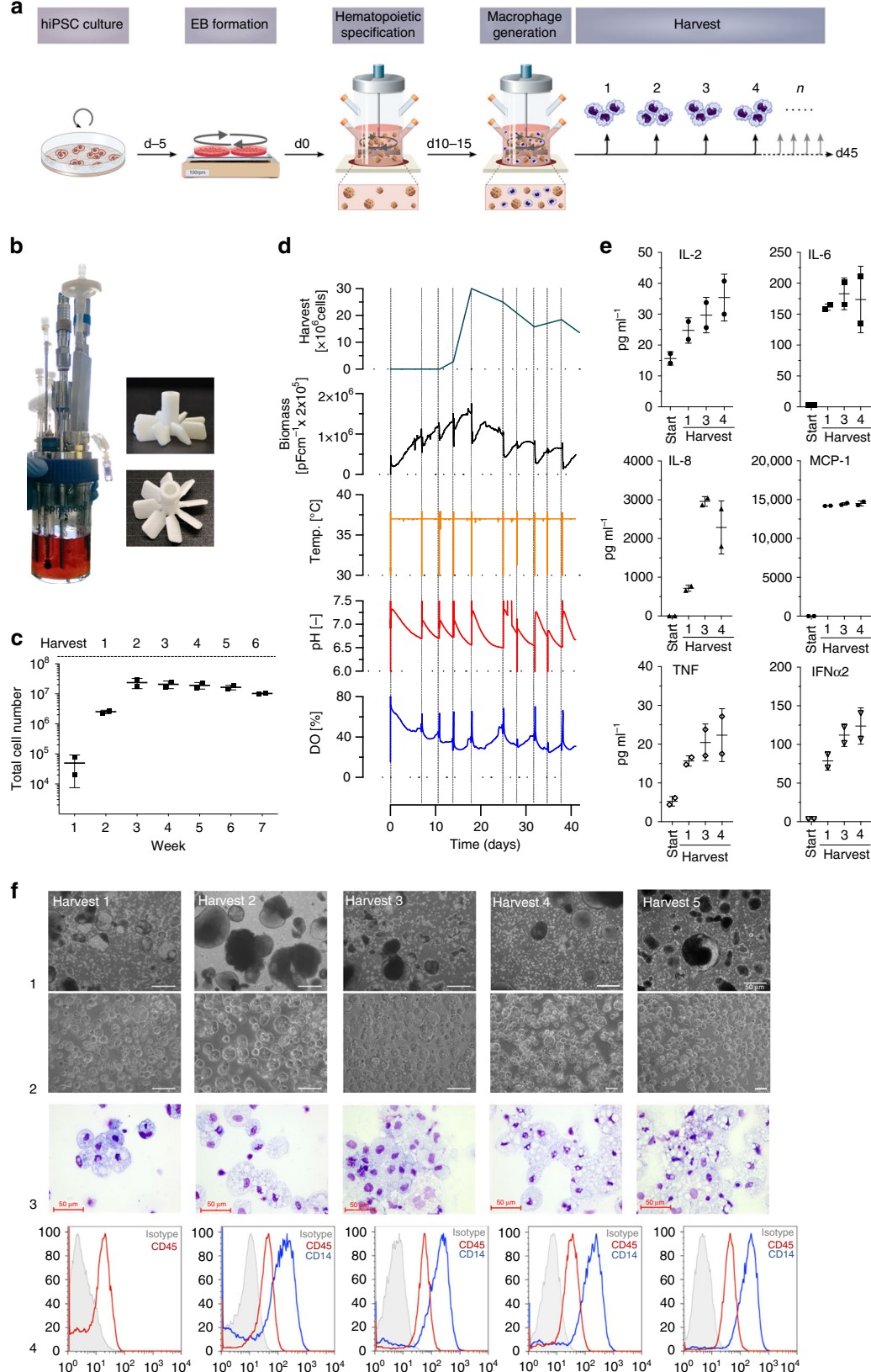

of the insulin-like growth factor axis. Interestingly, gene set enrichment analysis (GSEA) revealed a clear enrichment of genes associated with yolk sac-derived macrophages in iPSC-Mac (Fig. 2f), including the DNA-binding protein inhibitor ID1, transforming growth factor-beta 2 (*TGFB2*) and insulin-like growth factor 2 (*IGF2*) (Fig. 2g), suggesting a transcriptional phenotype of iPSC-Mac similar to primitive macrophages from the yolk sac.

Taken together, our iPSC-Mac show close transcriptional similarities to PBMC-Mac, however, with a distinct signature towards primitive macrophages.

**Fig. 1** Continuous generation of human iPSC-Mac in stirred tank bioreactors. **a** Scheme of hematopoietic differentiation of human iPSC in stirred tank bioreactors (DASbox system). **b** Representative pictures of DASbox bioreactor filled with floating MCFCs (left). Images of the 8-blade impeller (right). **c** Individual cell counts of viable macrophages produced in bioreactors (*n* = 2 of independent bioreactor runs, mean ± SD.). **d** Representative macrophage harvest counts (upper graph) and corresponding data from continuous process monitoring (biomass, temperature, pH, and dissolved oxygen (DO) level) for the entire cultivation phase of one 42-day bioreactor run. **e** Analysis of human cytokines in the medium from the bioreactor (technical duplicates, mean ± SD). **f** Representative light microscopy of macrophage generation analyzed at harvests 1–5. First row shows brightfield images of non-filtered medium samples from bioreactors (scale bar 500 μm). Second row shows brightfield images of freshly harvested macrophages separated from MCFCs by sedimentation (scale bar 50 μm). Third and fourth row show cytospin (scale bar 50 μm) and flow cytometric analysis of iPSC-Mac (gray line: respective isotype control, red line: CD45, blue line CD14)

**Antimicrobial activity of bioreactor-derived iPSC-Mac.** Next, bioreactor-derived iPSC-Mac were evaluated for their in vitro antimicrobial activity. Scanning electron microscopy (SEM) at different time points after incubation of iPSC-Mac with fluorescently labeled latex beads revealed typical changes in the overall cell morphology early after the stimulus and efficient phagocytic uptake of beads over time (Fig. 3a). IPSC-Mac also phagocytosed GFP-labeled *P. aeruginosa* (PAO1) at 37 °C in comparable efficiency to PBMC-Mac, whereas no phagocytosis was observed at 4 °C, confirming active phagocytosis (Fig. 3b and Supplementary Figure 3a). To gain insights into the ability of iPSC-Mac to remodel their transcriptome towards the signature of activated macrophages, we performed whole-transcriptome analysis of iPSC-Mac and PBMC-Mac before and after contact with *P. aeruginosa* (PAO1). Hierarchical cluster analysis of genes associated with inflammatory response (Fig. 3c) and innate immune response (Fig. 3d) showed marked upregulation of cytokines (e.g. *IL23A, TNF, IL1A/B, IL6*), chemokines (e.g. *CCL3, CCL4, CCL20, CXCL2, CXCL3*) and molecules involved in NFκB signaling in iPSC-Mac as well as PBMC-Mac in response to pathogen contact. Although both macrophage types showed characteristic transcriptional activation after stimulation, iPSC-Mac showed a more pronounced response (Fig. 3c, d; Supplementary Fig. 3b). This was further supported by analysis of >5-fold upregulated genes after pathogen contact showing 136 genes upregulated in iPSC-Mac compared to only 40 upregulated genes in PBMC-Mac (Fig. 3e). PBMC-Mac shared 29 (72.5%) of those upregulated genes, including important regulators of innate immunity, such as the immune-responsive gene 1 (*IRG1*), *IL6, CXCL1,2* and *3, IL1A* and *B, TNF*, with iPSC-Mac. Gene ontology enrichment analysis of genes only upregulated in PBMC-Mac revealed highest scores for GOs associated with activation of *IL10*, GM-CSF, *IL17*, and *IL12*, whereas genes upregulated in iPSC-Mac revealed enrichment in GOs associated with regulation of transcription, as well as acute inflammatory response. To investigate the quality of the antimicrobial response, we next analyzed the top 100 upregulated genes in iPSC-Mac or PBMC-Mac following pathogen challenge. Here, GO enrichment analysis for molecular function and biological processes, revealed similar GOs associated with chemokine and cytokine activity (Fig. 3f) as well as inflammatory response (Fig. 3g) in both cell types. Taken together, both, iPSC-Mac and PBMC-Mac demonstrate overall similar characteristic transcriptional changes after pathogen contact, but with a more pronounced response in iPSC-Mac.

**iPSC-Mac prevent respiratory infections by *P. aeruginosa*.** To evaluate the therapeutic efficacy of bioreactor-derived iPSC-Mac in vivo, we utilized the humanized mouse model C;129S4-*Rag2*[tm1.1Flv] *Csf1*[tm1(CSF1)Flv] *Csf2/Il3*[tm1.1(CSF2,IL3)Flv] *Thpo*[tm1.1(TPO)Flv] *Il2rg*[tm1.1Flv] Tg(SIRPA)1Flv/J (hIL-3/GM-CSF KI), an immunodeficient strain with impaired alveolar macrophage development and an established model to study pulmonary infections[19–21]. Given its frequent application in several murine infections models, including pulmonary infections, we used the *P. aeruginosa* strain PAO1 for our infection experiments[22,23].

We first established an acute pulmonary infection model in hIL-3/GM-CSF KI mice by intratracheal installation of PAO1. In line with the current animal welfare regulations, we were able to identify a suitable dose of *P. aeruginosa*, which leads to a prominent infection in mice represented by a disease score of 6–8 in hIL-3/GM-CSF KI mice (0: unaffected, 10: moribund) within 24 h post infection without leading to mortality of the animals. Applying the afore mentioned protocol, hIL-3/GM-CSF KI mice were infected with the *P. aeruginosa* strain PAO1 and co-administered with $4 \times 10^6$ bioreactor-derived, non-purified iPSC-Mac in the same instillation process (solely *PAO1* infected mice served as controls). The infection course was closely monitored and mice were killed 24 h post infection for end analysis (Fig. 4a).

Infected hIL-3/GM-CSF KI mice displayed clinical symptoms already 4 h post infection as indicated by an elevated disease score of 3.5 ± 0.9, which gradually increased over time up to 6.1 ± 0.4 after 24 h. In addition, a decrease in rectal temperature to 35.0 ± 0.3 °C and body weight loss of 2.6 ± 0.5 g was observed in infected animals 24 h post infection (all mean ± s.e.m., *n* = 6). In contrast, animals receiving a simultaneous pulmonary iPSC-Mac transplantation (PiMT) showed only very mild symptoms of infection (Fig. 4b, c and Supplementary Fig. 4a). This was in line with normal rectal temperature of 37.1 ± 0.2 °C, a very low disease score of 0.6 ± 0.2 and only little loss of body weight of −0.89 ± 0.2 g 24 h post infection in PiMT-treated animals (all mean ± s.e.m., *n* = 6) (Fig. 4b, c). Similar beneficial effects of PiMT were demonstrated by head-out body-plethysmography to measure lung function[24]. Whereas infected animals showed a decrease in tidal volume as well as expiratory and inspiratory time and an increase in breathing rate, animals from the infected + PiMT group showed normal values for all parameters analyzed (Fig. 4d and Supplementary Fig. 4b). As a consequence of PiMT, transplanted mice showed significantly reduced bacterial numbers in the lung 24 h post infection compared to their non-transplanted controls (Fig. 4e). Moreover, erythrocytes were present only in bronchoalveolar lavage fluid (BALF) from infected mice not receiving PiMT (Fig. 4f). Lung inflammation was furthermore indicated by an increase in murine GR1+ granulocytes in the BALF and lungs of infected animals, but not infected + PiMT or control mice (Fig. 4g and Supplementary Figure 4c). In addition, histological evaluation revealed massive granulocyte infiltration, severe hemorrhage and alveolar edema in infected mice (score: 13.7 ± 0.3), whereas lungs from infected + PiMT animals showed merely slight changes (score: 2.0 ± 1.2, both mean ± s.e.m., *n* = 3) (Fig. 4h and Supplementary Fig. 4d). Reduced inflammation in infected + PiMT animals was associated with the detection of hCD45+ cells in lung and BALF as well as presence of macrophages in histological sections of the right lung (Fig. 4i and Supplementary Fig. 4e).

**PiMT rescues mice from severe respiratory infections.** After demonstrating the efficacy of iPSC-Mac in simultaneous infection experiments, we evaluated a clinically more relevant therapeutic PiMT treatment approach. In these experiments, hIL-3/GM-CSF

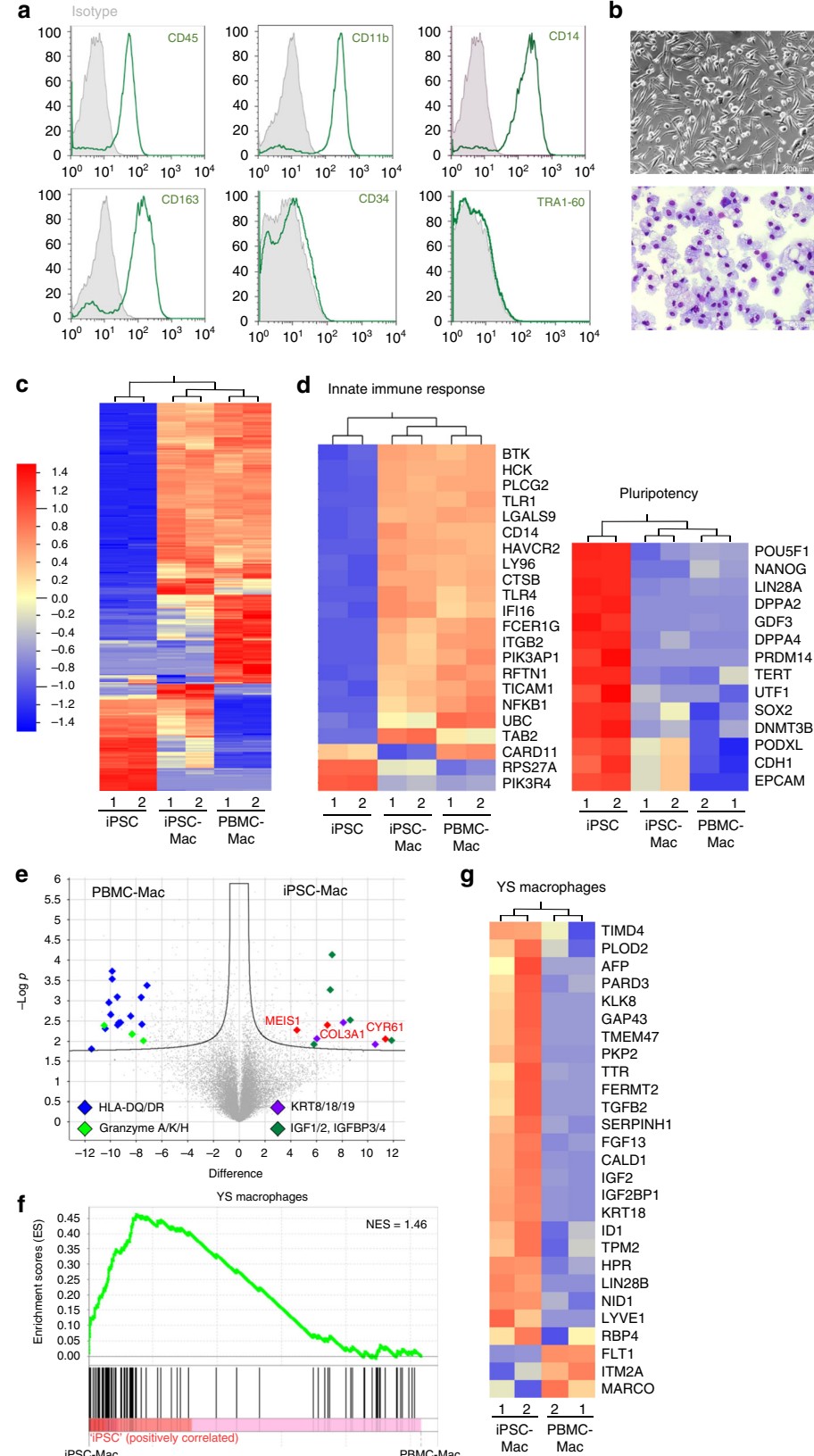

KI mice were infected intra-pulmonary with *P. aeruginosa* and closely monitored for 3–4 h until first disease symptoms developed (determined by a disease score ≥5). Subsequently, mice received $4 \times 10^6$ iPSC-Mac (infected + PiMT) only after the infection-related symptoms were readily manifested. Instead of

PiMT, infected control mice received PBS only (infected) (Fig. 5a).

In our infection and treatment schedule, therapeutic PiMT-treated mice showed a decrease in the disease score as well as a normalization of rectal temperature comparable to non-infected

**Fig. 2** Characterization of iPSC-Mac derived from stirred tank bioreactors. **a** Representative flow cytometric analysis of iPSC-Mac derived from bioreactor differentiation (pre-gated on viable cells in forward and sidewards scatter (FSC/SSC) analysis, gray line: respective isotype control, green line: cell surface marker). **b** Representative pictures of either brightfield (upper image, scale bar 200 μm) or cytospin preparations (lower image, scale bar 100 μm) of terminally differentiated iPSC-Mac derived from the bioreactor. **c–g** Transcriptome analysis of iPSC, iPSC-derived macrophages (iPSC-Mac) and PBMC-derived macrophages (PBMC-Mac) ($n = 2$, biological replicates). **c** Unbiased, hierarchical heatmap clustering. **d** Heatmap of differentially regulated genes ($p < 0.001$) associated with activation of innate immune response (GO:0002218 left) and pluripotency-associated genes (right). **e** Volcano plot showing differentially expressed genes between PBMC-Mac and iPSC-Mac (FDR = 0.5). **f** Gene set enrichment analysis (GSEA) from iPSC-Mac versus PBMC-Mac shows enrichment of genes related to yolk sac-derived macrophages. The gene set was derived from>5-fold upregulated genes of murine yolk sac (YS)-derived macrophages compared to bone marrow-derived macrophages. NES – normalized enrichment score. **g** Heatmap of >5-fold regulated genes associated with YS macrophages shows predominantly upregulation in iPSC-Mac compared to PBMC-Mac

animals already within 4–8 h post treatment. In contrast, infected mice receiving PBS showed pronounced disease progression over time (Fig. 5b). Of note, 24 h post infection, infected mice showed marked disease symptoms, whereas animals that received a therapeutic PiMT showed a significantly reduced disease score ($1.8 \pm 0.2$ infected + PiMT vs $8.1 \pm 0.2$ for infected animals, mean ± s.e.m., $n = 3$) (Fig. 5b, c). In addition, the elevated disease score in infected mice was associated with clearly restricted activity when compared to control animals and mice receiving the therapeutic PiMT (representative trajectory shown for one animal each, Fig. 5d). Efficiency of therapeutic PiMT was further documented by normalized rectal temperature and a tendency towards maintained body weight values 24 h post infection (Fig. 5e, f). Moreover, a profound reduction in lung bacterial numbers in infected + PiMT mice was observed (Fig. 5g). Recapitulating our findings in the simultaneous transplant model, BALF of animals from the infected + PiMT group showed reduced erythrocyte levels compared to infected, non-transplanted controls (Fig. 5h). This observation was accompanied by the detection of important pro-inflammatory human cytokines, such as hIL6, hIL8, hINFα2, hMCP-1, and TNF only in animals receiving a PiMT, whereas human cytokines could not be detected in control or infected only animals (Fig. 5i and Supplementary Fig. 5a). In contrast, levels for murine pro-inflammatory cytokines (mIL1b, mIL6, mMCP-1, mTNF) were highly elevated in infected only animals, whereas almost normal levels could be detected in PiMT-treated animals (Supplementary Fig. 5b). Inflammation was furthermore assessed by lung histology sections. Here, infected mice showed several areas of massive granulocytic infiltration, severe hemorrhage and alveolar edema, which were barely detectable in mice treated with iPSC-Mac from the bioreactor (Fig. 5j).

## Discussion

In the present study, we introduce an immunotherapy approach using iPSC-Mac to enhance innate pulmonary immunity and combat bacterial infections. As clinically relevant numbers of macrophages can hardly be produced from somatic cell sources, we demonstrate the use of human iPSC and their scalable differentiation in industry-compatible bioreactors for generating substantial quantities of functional macrophages.

The strong and rapid therapeutic effect observed in our proof-of-concept study is of particular importance for a broad applicability of iPSC-Mac to target bacterial infections and quick clinical implementation. We evaluated the antimicrobial activity of iPSC-Mac against *P. aeruginosa*. However, a phagocyte-based immunotherapy approach might allow for a broad spectrum of applications in a number of different infection scenarios caused by other Gram-negative or -positive bacteria, such as *Streptococcus pneumoniae* or *Staphylococcus aureus*. Moreover, this approach might also be used to target pathogens associated with implant infections, which meanwhile pose an immense health and economical problem. This being said, new treatment options

are of specific clinical relevance considering the increasing numbers of pathogens, which are refractory to standard or reserve antibiotic therapy[25–27].

As a future clinical prerequisite, iPSC-Mac generated with our bioreactor platform represent a highly pure population without implementation of further FACS or MACS-based enrichment strategies and shared phenotypic, transcriptional and functional hallmarks with PBMC-Mac. Despite these similarities, iPSC-Mac exhibited a stronger response to pathogen stimulation as demonstrated by whole transcriptome analysis, further favoring their application as a cellular therapeutic for the treatment of bacterial infections.

For our immunotherapy approach, we employed an acute infection model using hIL-3/GM-CSF KI mice, an immunodeficient mouse strain with impaired alveolar macrophages development[19]. In this model, treatment of acute *P. aeruginosa*-induced pneumonia either by simultaneous or, even more important, therapeutic PiMT was highly effective within a short time frame. These proof-of-concept experiments were performed with a high cell dose of $4 \times 10^6$ iPSC-Mac per single intra-pulmonary application. However, even with this maximum cell dose, no apparent adverse events were observed in either treatment scenario. This suggests that transplantation of iPSC-Mac is well tolerated and safe, concurring with observations from pulmonary macrophage transplantation studies using macrophages from different stem cell sources[28–31]. Nevertheless, more studies are needed to address potential immunological side effects induced by the transplanted macrophages. Similarly, longer follow-up, detailed posology as well as intense preclinical safety and toxicology studies will be required prior to clinical translation.

Considering a body weight of 25 g per mouse, clinical translation of iPSC-Mac for the treatment of respiratory infections would require a maximal cell dose of $1 \times 10^{10}$ iPSC-Mac for a complete lung of a 60 kg patient. Even without further process optimization, this would translate to a 40–60 l production scale, which is in principle feasible with current bioreactor technologies[14]. Continuous process monitoring of our bioreactor-based differentiation revealed a substantial increase in biomass especially in the first 2–3 days of the repeated batch culture. This was followed by partial recovery of dissolved oxygen levels and a profound drop in pH in the last days before medium refreshment. These observations suggest the presence of process-limiting factors and highlight the potential for further process optimization. This is achievable by cultivation at higher cell densities combined with the application of perfusion systems and feedback control of oxygen, pH, and other process parameters, which can multiply cell yields of hPSC and their progenies[32].

Given the impressive in vivo functionality of iPSC-Mac in the acute infection model, a potentially important extension of this work would be to evaluate the therapeutic efficacy of a macrophage-based immunotherapy in chronic infections with *P. aeruginosa*, as frequently observed in patients suffering from chronic obstructive pulmonary disease (COPD)[33] or cystic fibrosis (CF)[34]. As previously shown for endogenous alveolar

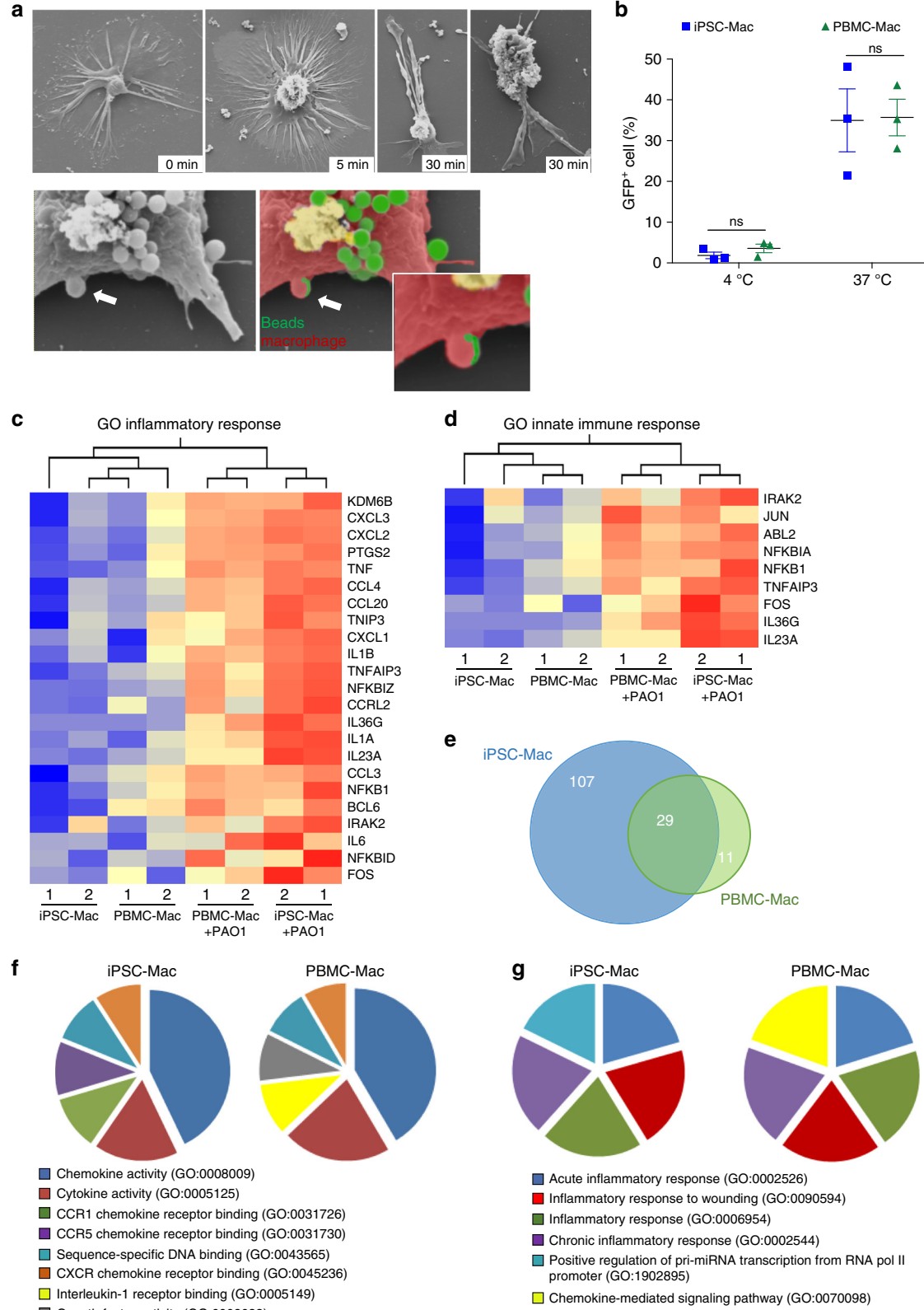

**Fig. 3** Antimicrobial activity of iPSC-Mac generated in stirred tank bioreactors. **a** Scanning raster electron microscopy depicting phagocytosis of latex beads (green) by iPSC-Mac (red) at different time points. **b** Rate of phagocytosis by terminally differentiated iPSC-Mac (blue) and PBMC-Mac (green) after 2 h of incubation with GFP-labeled *P. aeruginosa* (PAO1) at 4 or 37 °C ($n = 3$ of biological replicates, mean ± s.e.m, two-way ANOVA with Sidak's multiple comparisons test, ns denotes not significant, representative data and gating strategy provided in Supplementary Figure 3a). **c**, **d** Heatmap of commonly regulated genes in iPSC-Mac and PBMC-Mac associated with (**c**) inflammatory response (GO:0006954; $\sigma/\sigma_{max} = 0.13$, $P < 0.05$) and (**d**) innate immune response (GO:0045087, $\sigma/\sigma_{max} = 0.13$, $P < 0.05$) after 1 h exposure to PAO1. **e** Venn-diagram of >5-fold upregulated genes in iPSC-Mac (blue) and PBMC-Mac (green) after pathogen contact. **f**, **g** Top-ranked GOs associated with **f** biological processes and **g** molecular function (according to human gene atlas) of top 100 upregulated genes after 1 h pathogen contact (without pre-selection, EnrichR)

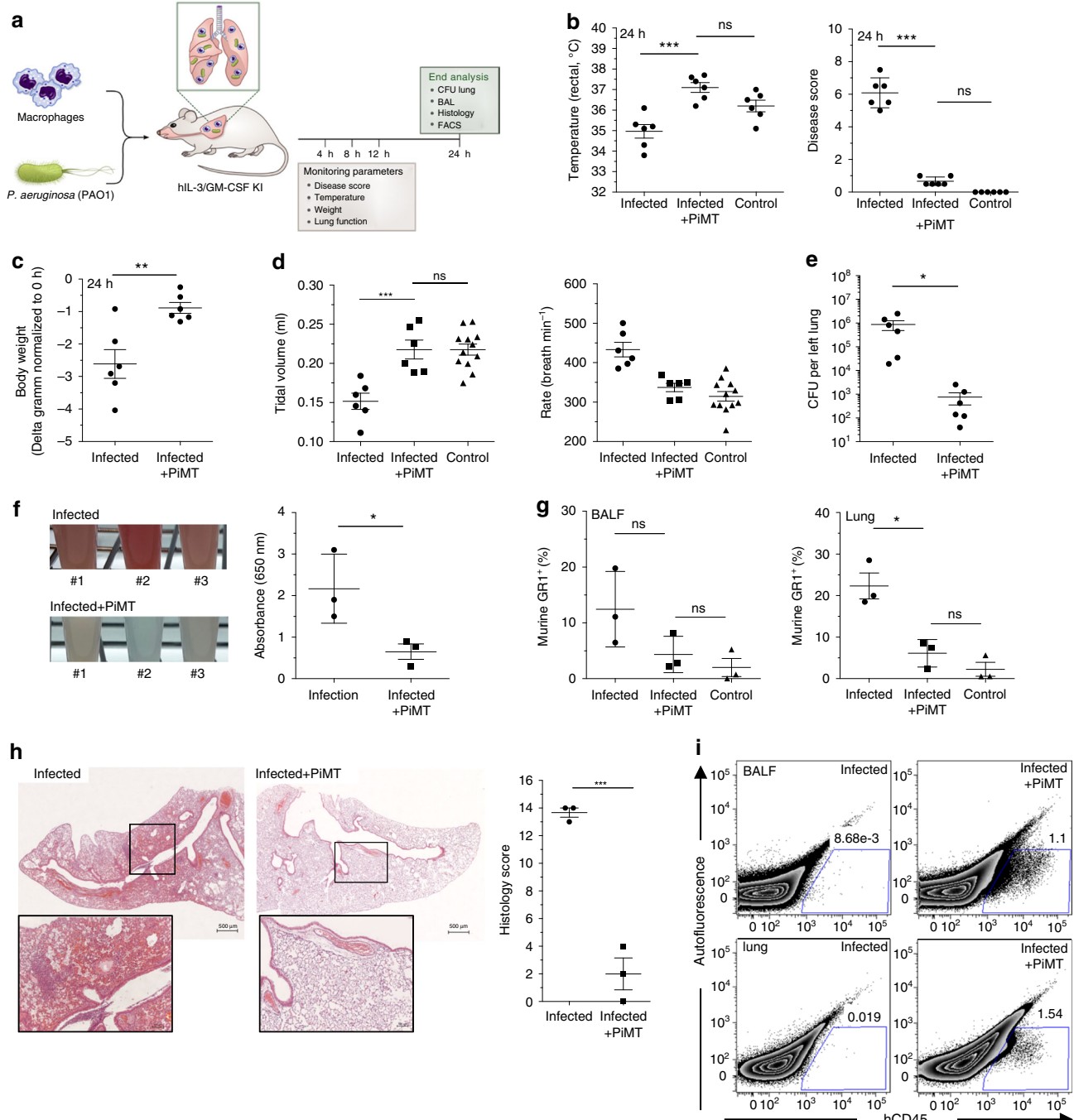

**Fig. 4** Pulmonary infection and simultaneous macrophage transplantation in hIL-3/GM-CSF KI mice. **a** Scheme of pulmonary transfer of *P. aeruginosa* (PAO1) and simultaneous transplantation of iPSC-Mac (PiMT) into hIL-3/GM-CSF KI mice. **b** Rectal temperature and disease score 24 h post infection of hIL-3/GM-CSF KI mice infected with PAO1 (infected) or infected and transplanted (infected + PiMT). Disease score and temperatures measured before the experiment served as control values ($n = 6$ animals/group, mean ± s.e.m). **c** Change in body weight after 24 h. Values are normalized to the respective weights before infection ($n = 6$ animals/group, mean ± s.e.m). **d** Lung function measured by head-out body plethysmography. Values measured pre infection served as control values ($n = 6$ animals/infected and infected + PiMT groups, $n = 12$ animals for control values, mean ± s.e.m). **e** Colony forming units (CFU) of PAO1 per left lung after 24 h ($n = 6$ animals/group, mean ± s.e.m). **f** Images of bronchio-alveolar lavage fluid (BALF) samples and BALF absorbance at 650 nm ($n = 3$ animals/group, mean ± s.e.m). **g** Flow cytometric analysis of BALF and lung. Percentage of mouse granulocytes (determined as GR1+ cells) in BALF and Lung ($n = 3$ animals/group, mean ± s.e.m, exemplarily data and gating strategy provided in Supplementary Figure 4c). **h** Right lung histology. Left: Two representative images of infected (left) and infected + PiMT (right) mice. Scale bars: 500 μm upper row and 100 μm lower row. Right: histological scoring ($n = 3$ animals/group, mean ± s.e.m). **i** Representative diagram of BALF and lung of one animal per group stained with hCD45. (*$p < 0.05$, **$p < 0.01$, ***$p < 0.001$, ****$p < 0.0001$, ns denotes not significant; statistical significances were assessed using one-way ANOVA with Dunnett's multiple comparisons test (**b**, **d**, **g**) or Student's *t*-test (**c**, **e**, **f**, **h**))

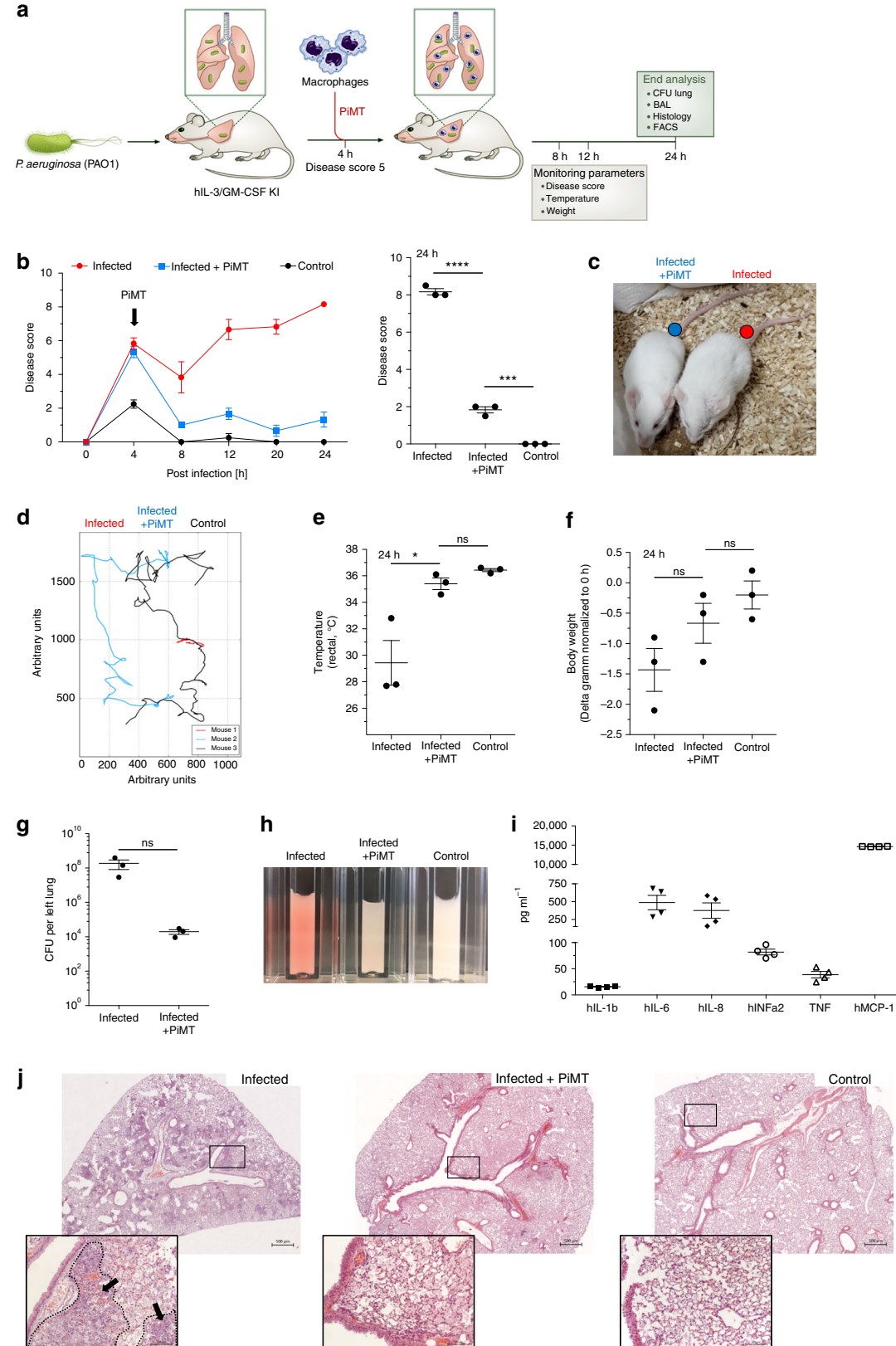

macrophages[35], iPSC-Mac might be induced towards a vaccine-induced macrophage (ViM) phenotype, which upon intra-pulmonary administration could protect mice from *P. aeruginosa* infections. This might be of particular interest in life-threatening, chemotherapy associated infections. Whereas in our "*protective*" regimen, we have used the simultaneous administration of iPSC-Mac and *P. aeruginosa*, administration of iPSC-derived ViM prior the infection would provide additional insights in the protective properties. Along this line, further long-term studies are needed to assess potential long-term pulmonary engraftment and sustained functionality of the transplanted iPSC-Mac. In our transplantation scenarios, iPSC-Mac were still

**Fig. 5** Pulmonary infection and therapeutic transplantation of iPSC-Mac in hIL-3/GM-CSF KI mice. **a** Scheme of pulmonary infection with *P. aeruginosa* (PAO1) and therapeutic transplantation of iPSC-Mac (PiMT) derived from bioreactors into hIL-3/GM-CSF KI mice. **b** Disease score of hIL-3/GM-CSF KI mice infected with PAO1 (infected, red), infected and transplanted (infected + PiMT, blue) and control mice receiving PBS twice (control, black). Left: Disease score over time. Right: Disease score 24 h post infection ($n = 3$ animals/group, mean ± s.e.m). **c** Infected mice (red dot) and infected + PiMT mice (blue dot) 24 h after infection. **d** Representative trajectory of mouse activity 24 h post infection for hIL-3/GM-CSF KI mice infected with PAO1 (infected, red), infected and transplanted (infected + PiMT, blue) and control mice receiving PBS twice (control, black) analyzed by video documentation and manual tracking. **e** Rectal temperature analyzed 24 h post infection ($n = 3$ animals/group, mean ± s.e.m). **f** Change in body weight after 24 h. Values are normalized to the respective weights before infection ($n = 3$ animals/group, mean ± s.e.m). **g** Colony forming units (CFU) of PAO1 per lung after 24 h ($n = 3$ animals/group, mean ± s.e.m). **h** Images of BALF samples. **i** Levels of human cytokines in BALF of infected + PiMT mice ($n = 4$: 2 animals and 2 technical duplicates, mean ± s.e.m). **j** Lung histology images of infected (left), infected + PiMT (middle) and control (right) mice. Scale bars: 500 μm upper row and 100 μm lower row. (*$p < 0.05$, **$p < 0.01$, ***$p < 0.001$, ****$p < 0.0001$, ns denotes not significant; statistical significances were assed using one-way ANOVA with Dunnett's multiple comparisons test (**b**, **e**, **f**) or Student's *t*-test (**g**))

detected at 24 h, however, later time points were not investigated. Earlier studies using the same mouse model demonstrated long-term pulmonary engraftment of cord blood- or iPSC-derived macrophages, whereas no cells could be detected in other organs associated with homing of macrophages[28,29]. Although the open lung microenvironment may serve as an instructive signal for the local engraftment of cells, the ancestry of the individual macrophages subset also has to be considered in more detail. Previous studies suggested a primitive fingerprint of murine and human iPSC-derived macrophages[30,36,37], which is further supported by our microarray analysis. Although our cells seem to have a primitive origin, previous studies revealed a similar adaption potential of iPSC-Mac and CD34+-Mac, highlighting a similar tissue plasticity[28]. Clearly, further studies are needed to evaluate whether iPSC-Mac can also provide a long-term antimicrobial defense and integrate into the lung environment after the infection is solved.

Taken together, we here introduce an immunotherapy approach employing iPSC-Mac to target bacterial airway infections. To translate this concept to clinical application, either allogeneic or genetically modified universal iPSC[38] may be employed to produce suitable macrophages in a cost-effective way by industrial scale bioreactor systems as an off-the-shelf product. While our proof-of-concept study is done in an acute infection mouse model, a first application of an iPSC-Mac-based therapy (iMATH) approach in humans may be envisioned in CF patients suffering from chronic *P. aeruginosa* infections. Given the defined clinical background of such patients, first-in-human-studies may be performed in a homogenous patient population before this approach is translated to patients suffering from acute and life-threatening respiratory infections. Further development of this technology, including process upscaling and the generation of additional hematopoietic cell types as well as assessment in other preclinical models may allow for innovative cell-based treatment strategies for a wide variety of diseases.

## Methods

**iPSC cultivation**. Human iPSC (hCD34iPSC16) have been previously generated from mobilized peripheral blood CD34+ cells[16] and were culture on irradiated murine embryonic fibroblasts (MEF) in human iPSC medium (knockout DMEM, 20% knockout serum replacement, 1 mM L-glutamine, 1% NEAA, 1% penicillin/streptomycin (all Invitrogen), 0.1 mM β-mercaptoethanol (Sigma-Aldrich), supplemented with 10 ng per mL bFGF (PeproTech). Cells were splitted every 7 days using 0.2% collagenase IV for 10–15 min. Basic-FGF was omitted from the maintenance medium for the last 3–5 days prior to EB formation. Human iPSC cultures have been regularly screened for mycoplasma contamination by PCR.

**Hematopoietic differentiation in suspension culture**. To induce hematopoietic differentiation in suspension, ~80% confluent hiPSC cultures on day 7–10 after passaging were harvested using 0.2% collagenase IV for 10–15 min. EBs were formed by pooling collected cell clusters from 3 wells of a 6-well plate per 1 well of a 6-well suspension plate in 3 mL of hiPSC medium without bFGF, but supplemented with 10 μM Y-27632 (Tocris). After 5 days incubation on an orbital shaker

(Celltron, Infors HT) at 85 revolutions per minute (rpm), largest EBs (>200 μm diameter) were selected manually with a binocular or by sedimentation properties (sedimentation for 10–15 min) and transferred to a new 6-well plate containing 3 mL differentiation medium I (X-VIVO 15 (Lonza), 1% penicillin-streptomycin (Life Technologies), 1 mM L-glutamine, and 0.05 mM β-mercaptoethanol (Sigma-Aldrich)) supplemented with 25 ng per mL IL3 and 50 ng per mL M-CSF. Medium was refreshed every 6–7 days. From day 10 to 15 onwards, iPSC-Mac were collected from the medium once a week.

**Hematopoietic differentiation in stirred tank bioreactors**. The bioreactor (DASbox Mini bioreactor system, Eppendorf) was setup and calibrated as previously described[39]. In brief, the 250 mL glass vessel was equipped with an 8-blade impeller (60° pitch) and probes for online monitoring of biomass (Aber Instruments), pH, DO as well as control of temperature. Calibration was performed in 120 mL chemically defined X-VIVO 15 (Lonza).

For hematopoietic differentiation in the bioreactor, iPSC were expanded to 20 6-well plates, maintained in human iPSC-medium for 4–7 days and then cultivated for 3 days in the absence of bFGF. EB formation was performed equivalent to suspension cultures. After 5 days, EBs were selected by sedimentation properties (sedimentation for 10–15 min) to exclude debris, single cells as well as small cell clusters and subsequently transferred to equilibrated bioreactors. Cells were cultivated in differentiation medium I at 37 °C with constant headspace-gassing at 3 L per hour (21% $O_2$; 5% $CO_2$) and stirring at 50 rpm. To monitor MCFCs integrity and macrophages formation, 1 mL samples were collected 1–2 times per week via the sampling port without interrupting the culture process. Differentiation medium I (X-VIVO 15 (Lonza), 1% penicillin-streptomycin (Life Technologies), 1 mM L-glutamine and 0.05 mM β-mercaptoethanol (Sigma-Aldrich)) supplemented with 25 ng per mL IL3 and 50 ng per mL M-CSF) was manually replaced every 6–7 days with optional fed of 20 mL after 3–4 days. Macrophages were collected weekly by separation from MCFC via sedimentation (4–5 min) and subsequent filtering of the medium through a 100 μm strainer (PluriSelect). Retained MCFCs were returned to the bioreactor. Macrophages were collected from filtered medium via centrifugation at 300×g for 4 min.

Data from online monitoring were processed using Microsoft Excel 2016 and GraphPad Prism 6. Supernatant was analysed for concentration of glucose and lactate using YSI 2700 select biochemistry analyser, for osmolarity using Osmomat 300 (Gonotec) and for concentration of lactate dehydrogenase according to manufacturer's instruction (MAK066, Sigma) using a microplate reader (Paradigm, Beckman Coulter).

**Terminal differentiation**. For further maturation, cells freshly collected from MCFCs were cultured in differentiation medium II (RPMI1640 medium supplemented with 10% fetal calf serum (FCS), 2 mM L-glutamine, 1% penicillin-streptomycin) containing 50 ng per mL hM-CSF for at least 7 days.

**Generation of PBMC-derived macrophages**. Donation of peripheral blood and isolation of peripheral blood mononuclear cells (PBMC) was approved by the local ethical committee (Ethics Committee Hannover Medical School). All healthy donors gave written informed consent. PBMC were isolated from the peripheral blood of healthy volunteers by gradient centrifugation using Biocoll Separating Solution (40 min, 400×g; Biochrome, Billerica, MA). Subsequently, cells were cultured in RPMI1640 medium supplemented with 10% fetal calf serum, 2 mM L-glutamine, 1% penicillin-streptomycin (all Invitrogen), and hIL-3 and hM-CSF (50 ng per mL each, PeproTech) for 1 week. After this, PBMC-Mac were cultivated for further 3–4 days in differentiation medium containing 50 ng per mL M-CSF only.

**Flow cytometry**. Flow cytometric analysis of myeloid cells was performed as described[16,40]. For macrophages, PBS supplemented with 10% FCS was used to prevent unspecific binding. Cells were stained with 0.5–1 μl of the respective antibody and analyzed with a FACScalibur cytometer (Beckton & Dickinson, Heidelberg, Germany) and analyzed with FlowJo software (TreeStar, Ashland, OR).

Antibodies were purchased from eBioscience: hTRA-1-60-PE (Cat-No: 12-8863-80), hCD11b-APC (Cat-No: 17-0118-41), hCD14-PE (Cat-No: 12-0149-42), hCD163-APC (Cat-No: 17-1639-41), hCD16-FITC (Cat-No: 11-0168-41), hCD34-FITC (Cat-No: 11-0349-41) and isotype-controls: mouse-IgG1a-PE (Cat-No: 12-4714-41), FITC (Cat-No: 11-4714-41) or APC (Cat-No: 17-4714-41), and rat-IgG2a-PE (Cat-No: 12-4321-81). Antibodies from Biolegend San Diego, CA, United States: hCD86-APC (Cat-No: 305411), hCD66b-FITC (Cat-No: 305104), or hCD45-PE (Cat-No: 304007).

For flow-cytometric analysis of mouse lung and BALF, samples were fixed using 4%PFA. After this, samples were incubated with 1 μl fc receptor blocking antibodies (CD16/CD32, eBioscience, Cat-No: 14-0161-81) for 20 min to prevent unspecific binding and finally stained with 1 μl of the respective antibody for 1 h at 4 °C in the dark. Used antibodies were purchased from Biolegend San Diego, CA, United States: hCD45-PeCy7, and eBioscience: mGR1-eFluor450. Samples were analyzed at a BD LSR II Flow Cytometer (Beckton & Dickinson, Heidelberg, Germany) and analyzed with FlowJo software (TreeStar, Ashland, OR).

**Cytospin preparation.** A total of 20,000–50,000 cells were spun on glass slides at 600×$g$ for 7 min and stained for 5 min in 0.25% May-Grünwald and 20 min in 0.4% Giemsa stain modified solution (Sigma).

**Phagocytosis assays.** The phagocytic activity of iPSC-Mac derived from suspension or adherent cultures and terminally differentiated on tissue-culture plates was assessed by flow cytometry. Therefore, $1 \times 10^5$ cells were incubated with pHrodo™ Red *E. coli* BioParticles® Conjugate (MolecularProbes/Thermo Fisher Scientific, Schwerte, Germany) or medium for 2 h at 37 °C or 4 °C as a negative control. After incubation, the cells were put on ice for 10 min. Analysis was performed using a Beckman Coulter FC500 flow cytometer.

The functional capacity of iPSC-Mac and PBMC-Mac to phagocytose vital *P. aeruginosa* was assessed using GFP-PAO1 (wild-type *P. aeruginosa* PAO1 tagged with green fluorescent protein (GFP) by Tn*7* transformation[41], (kindly provided by Thomas Bjarnsholt, University of Copenhagen) and compared to PBMC-derived Macrophages (PBMC-Mac). For the phagocytosis assay, $1 \times 10^5$ iPSC-Mac were incubated for 2 h with $6 \times 10^5$ CFU GFP-PAO1 at 37 °C or 4 °C as a negative control. Medium controls were treated similar. After incubation, cells were put on ice for 10 min and subsequently fixed with 2% paraformaldehyde (PFA) solution for 30 min. Analysis was performed using a Beckman Coulter FC500 flow cytometer.

**Electron microscopy.** Macrophages were grown on 1 cm diameter round coverslips. Latex beads (1 μm diameter) were added to cells at 4 °C and adhesion of latex beads to the surface of the macrophages was allowed for 5 min. Following samples were washed with cold PBS and warmed up with fresh culture medium to 37 °C. Phagocytosis was allowed for up to 1 h and samples were fixed at different time points using 1.5% Paraformaldehyde, 1.5% Glutaraldehyde in 150 mM HEPES, pH 7.35. Samples were then dehydrated using an increasing methanol series. Critical point drying was performed using a CPD030 critical point dryer (Balzers, Lichtenstein) following manufacturer instructions. Coverslips were then sputtered with gold (Sem Coating System, Polarion) and SEM was carried out using a Philips SEM 505 (Eindhoven, The Netherlands).

**Bacterial culture.** For experiments *P. aeruginosa* laboratory strain *PAO1*[22] was taken from a stock culture kept at −80 °C and grown in Luria Broth (LB) overnight. After washing with sterile PBS the desired infectious dose was extrapolated from a standard growth curve. For the determination of the actual dosage, inoculates were serially plated on LB agar plates via the drop-plate method[42] and CFU determined after 16–18 h incubation at 37 °C.

**Collection of microarray samples.** Terminally differentiated human iPSC-Mac or human PBMC-Mac were seeded on 24-well plates (500,000 cells/well) and cultured overnight. On the next day, cells were washed three times with PBS. Subsequently, *P. aeruginosa* laboratory strain PAO1 in RPMI medium without antibiotics (MOI10) was centrifuged onto the cells (600×$g$) and incubated at 37 °C. Cells with medium only served as non-infected controls. After 1 h cells were de-attached, washed and resuspended in RNA lysis buffer. RNA isolation was performed with RNAeasy micro Kit (Quiagen) according to manufacturer's instructions. Human iPSC samples were obtained after sorting of iPSC for TRA-1-60$^+$ to separate iPSC from feeder cells.

**Microarray experiments.** The Microarray utilized in this study represents a refined version of the Whole Human Genome Oligo Microarray 4 × 44 K v2 (Design ID 026652, Agilent Technologies), called '054261On1M' (Design ID 066335) developed in the Research Core Unit Transcriptomics (RCUT) of Hannover Medical School. Microarray design was created at Agilent's eArray portal using a 1 × 1 M design format for mRNA expression as template. All non-control probes of design ID 026652 have been printed five times within a region comprising a total of 181,560 Features (170 columns × 1068 rows). Four of such regions were placed within one 1M region giving rise to four microarray fields per

slide to be hybridized individually (Customer Specified Feature Layout). Control probes required for proper Feature Extraction software operation were determined and placed automatically by eArray using recommended default settings.

A quantity of 30 ng of total RNA was used to prepare aminoallyl-UTP-modified (aaUTP) cRNA (Amino Allyl MessageAmp™ II Kit; #AM1753; Life Technologies) as directed by the company (applying one-round of amplification). The labeling of aaUTP-cRNA was performed by use of Alexa Fluor 555 Reactive Dye (#A32756; LifeTechnologies).

cRNA fragmentation, hybridization, and washing steps were carried out as recommended in the 'One-Color Microarray-Based Gene Expression Analysis Protocol V5.7', except that 500 ng of each fluorescently labeled cRNA population were used for hybridization.

Slides were scanned on the Agilent Micro Array Scanner G2565CA (pixel resolution 3 μm, bit depth 20). Data extraction was performed with the 'Feature Extraction Software V10.7.3.1' using the extraction protocol file 'GE1_107_Sep09.xml', except that 'Multiplicative detrending' algorithm was inactivated.

Measurements of on-chip replicates (quintuplicates) were averaged using the geometric mean of processed intensity values of the green channel, 'gProcessedSignal' (gPS) to retrieve one resulting value per unique non-control probe. Single features were excluded from averaging, if they (i) were manually flagged, (ii) were identified as Outliers by the Feature Extraction Software, (iii) lie outside the interval of '1.42 × interquartile range' regarding the normalized gPS distribution of the respective on-chip replicate population, or (iv) showed a coefficient of variation of pixel intensities per Feature that exceeded 0.5.

Averaged gPS values were normalized by quantile normalization approach first. Subsequently, values were additionally processed by global linear scaling: All gPS values of one sample were multiplied by an array-specific scaling factor. This factor was calculated by dividing a 'reference 75th percentile value' (set as 1500 for the whole series) by the 75th percentile value of the particular Microarray to be normalized ('Array I' in the formula shown below). Accordingly, normalized gPS values for all samples (microarray datasets) were calculated by the following formula:

$$\text{normalized gPS}_{\text{Array i}} = \text{gPS}_{\text{Array i}} \times \left( 1500 \text{ per 75th percentile}_{\text{Array i}} \right).$$

Finally, a lower intensity threshold (surrogate value) was defined based on intensity distribution of negative control features. This value was fixed at 15 normalized gPS units. All of those measurements that fell below this intensity cutoff were substituted by the respective surrogate value of 15.

Normalized microarray data of all non-control features were imported into Omics Explorer software v3.2 (Qlucore) under default import settings for Agilent One Color mRNA Microarrays, except that any normalization option was deselected. Accordingly, data processing steps during import were: (1) log base 2 transformation, (2) baseline transformation to the median.

Heatmap clustering analysis and generation of GO-based heatmaps were performed in Omics Explorer. Top 100 upregulated genes were calculated using the RCUTAS tool (V1.7; Hannover Medical School) and processed using Venny 2.1 (http://bioinfogp.cnb.csic.es/tools/venny). Gene ontology analysis of biological processes, molecular function and cell type classification based on the human gene atlas were conducted using Enrichr (https://amp.pharm.mssm.edu/Enrichr). Gene set enrichment analysis was performed using GSEA (v3.0; Broad Institute). Gene set for YS macrophages was obtained from Takata et al. using >5-fold upregulated genes compared to BMDM[37] and converted using OrthoRetriever (v1.2; http://lighthouse.ucsf.edu/orthoretriever/). Volcano plots were visualized using Perseus (v.1.5.6.0; http://www.perseus-framework.org; FDR = 0.05; s0 = 0.1). Microarray data were deposited under accession number E-MTAB-5436 in the ArrayExpress database (www.ebi.ac.uk/arrayexpress).

**Cytokine secretion assays.** In order to analyze the secretion of human cytokines in bioreactor samples or human and murine cytokines BALF samples, Luminex® analysis with a Cytokine Human 14-Plex Panel or Mouse 5 -Plex Panel (Millipore, Schwalbach, Germany) was performed as described before[40]. Data were acquired on a Luminex-200 System and analyzed with the Xponent software v.3.0 (Life Technologies).

**Animal maintenance and infection.** All animal experiments were approved by the local animal welfare committee ("Niedersächsisches Landesamt für Verbraucherschutz und Lebensmittelsicherheit/ LAVES") and performed according to their guidelines.

hIL-3/GM-CSF KI mice (C;129S4-*Rag2*$^{tm1.1Flv}$ *Csf1*$^{tm1(CSF1)Flv}$ *Csf2/Il3*$^{tm1.1(CSF2,IL3)Flv}$ *Thpo*$^{tm1.1(TPO)Flv}$ *Il2rg*$^{tm1.1Flv}$ Tg(SIRPA)1Flv/J) mice[19] were obtained from the Jackson Laboratory and housed in the central animal facility of Hannover Medical School. Immunodeficient mice were maintained under pathogen free conditions in individually ventilated cages (IVC) with free access to food and water.

For the experiments we used male and female animals between 10 and 26 weeks of age. The animals were randomized between the groups to ensure a homogenous mixture of male/female animals and different ages. Only animals with a disease score of ≤1 were included in the study, no animals were excluded from the analysis. For infection with *P. aeruginosa* or pulmonary iPSC-Mac transplantation,

anesthetized (ketamine/midazolam) mice were instilled via the trachea after oral intubation. For the simultaneous infection experiments, iPSC-Mac ($4 \times 10^6$/animal) and PAO1 ($0.2 \times 10^5$ CFU/per animal) were resuspended in PBS and mixed in a total volume of 60 μl. For solely infected mice the same CFU was applied in 60 μl PBS. To avoid phagocytosis before instillation, cell/bacteria mixes were kept on ice all the time. For the therapeutic PiMT experiments, hIL-3/GM-CSF KI mice were infected with $0.3 \times 10^5$ CFU PAO1 (in a volume of 30 μl PBS) after anesthesia with ketamine/midazolam. Control mice received the same volume PBS. After 4 h, mice were anesthetized by isoflurane inhalation and the second instillation with 50 μl PBS or $4 \times 10^6$ iPSC-Mac in PBS was performed. Control mice again received the same volume PBS. iPSC-Mac were freshly harvested from suspension cultures after harvest ≥2. The disease score of the animals was assessed using a scoring matrix[43]. The investigators were not blinded during the experiments and analysis. After 24 h, animals were killed and end analysis was performed.

Trajectory of mouse activity was analyzed by video documentation and manual tracking. The video was recorded using a handheld mobile camera in sterile housing and stabilized by optical flow methods. Mice motion was then tracked with the help of manually annotated keypoints on the animals' heads for a time slot of 22 s. Total trajectory is given by accumulating all keypoints.

**Murine lung function**. Non-invasive head-out spirometry investigating 14 lung function parameters was performed on conscious restrained mice[24]. Mice were positioned in glass inserts, their breathing causes air to flow through a pneumo-tachograph. A pressure transducer creates an electrical signal, which is analysed using NOTOCORD HEM software (Version 4.2.0.241, Notocord Systems SAS, Croissy Sur Seine,France). The parameters of tidal volume (measured in mL), expiratory time, inspiratory time, time of inspiration + expiration, relaxation time (all measured in ms) and the flow at 50% of the expiratory tidal volume (EF50) were selected to characterize murine lung function during infection.

**Broncho-aleveolar lavage**. Bronchoalveolar lavage was performed by cannulating the murine trachea post-mortem. The right lung was rinsed with 1 mL of PBS for three times.

**Measurement of hemoglobin levels in BALF**. Fresh BALF was used for photo-metric analysis of hemoglobin. After this, BALF samples were centrifuged and supernatants were stored at −80 °C for Luminex analysis. Pellets were fixed and stained for flow cytometry.

**Lung bacterial numbers**. The right lungs of the killed mice were ligated, resected, and homogenized with a tissue homogenizer (Polytron PT 1200, Germany). Total bacterial numbers were assessed from serial dilutions of the homogenates, which were cultured on Luria–Bertani plates using the drop plate method[42].

**Histology**. At the dedicated time points the animals were killed. Right lungs were filled with OCT buffer and fixed in neutral buffered 4% PFA for 3 days at 4 °C. For control animals, left lungs were used. Tissues were trimmed according to the RITA-Guidelines[44], dehydrated (Shandon Hypercenter, XP) and subsequently embedded in paraffin (TES, Medite). Sections (2–3 μm thick, microtom Reichert-Jung 2030) were deparaffinized in xylene and H&E stained according to standard protocols. Blinded evaluation (Axioskop 40, Zeiss microscope) and histological scoring of the sections was performed as described before[45] by a trained pathologist.

**Statistics**. GraphPad Prism 6 and 7 was applied to perform unpaired Student's $T$-test or analysis of variance (ANOVA). For experiments $n > 3$, mean ± SD is given, for experiments $n < 3$, mean ± s.e.m. is plotted. For all experiments, individual datapoints are shown. Asterisks denote: *$P < 0.05$; **$P < 0.01$; ***$P < 0.001$; ****$P < 0.0001$.

**Study approval**. For the isolation of peripheral blood mononuclear cells: healthy donors gave written informed consent according to the local ethical committee at Hannover Medical School (Ethics Committee Hannover Medical School). For animal experiments: all animal experiments were approved by the animal welfare committee of lower Saxony ("Niedersächsisches Landesamt für Verbraucherschutz und Lebensmittelsicherheit/ LAVES") and performed according to their guidelines.

## Data availability

The authors declare that all data supporting the findings of this study are available within the article and its Supplementary Information files or from the corresponding author on reasonable request. Microarray data have been deposited in the ArrayExpress database under accession number E-MTAB-5436.

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

## Acknowledgements

The authors thank Doreen Lüttge, Theresa Buchegger, Yannik Dünow, Annika Franke, Gerhard Preiss, and Silke Hedtfeld (all Hannover Medical School) for assistance. Moreover, we are thankful to Christina Kropp and Caroline Halloin for assistance with bioreactor setup and maintenance, Janita Lührs, Laura Boge, and Sabine Schild (all Fraunhofer ITEM, Hannover) for performance of the phagocytosis as well as Andreas Pösch (Leibniz University Hannover) for help with image analysis. The authors also thank Oliver Dittrich-Breiholz and Heike Schneider (Research Core Unit Transcriptomics, Hannover Medical School) for performing microarray analysis as well as M. Ballmaier and C. Struckmann (Core Unit Cell Sorting, Hannover Medical School) for FACS. The authors thank Thomas Bjarnsholt, University of Copenhagen, for kindly providing the GFP-tagged PAO1. In addition, the authors thank B. Tümmler (Hannover Medical School) for discussions and scientific input to the infection experiments. The authors would like to thank Michael Morgan (Hannover Medical School) for review of the manuscript. The authors would also like to thank all donors for donating blood. This work was supported by grants from the Deutsche Forschungsgemeinschaft (Cluster of Excellence REBIRTH; Exc 62/1 to T.M., R.Z., U.M., A.S., and N.L. as well as DFG LA 3680/2-1 and ZW64/4-1), the Else-Kröner-Fresenius-Stiftung (EKFS 2013_A24 to T.M., EKFS; 2015_A92 to N.L. and EKFS; 2016_A146 to M.A.), the German Ministry for Education and Science (BMBF 13N12606,13N14086 and iMACnet 01EK1602A), Stem-BANCC (support from the Innovative Medicines Initiative joint undertaking under grant 115439-2, whose resources are composed of financial contribution from the European Union [FP7/2007-2013] and EFPIA companies' in-kind contribution), TECHNOBEAT (European Union H2020 grant 668724), the Joachim Herz Stiftung (to N.L., H.K., and M. A.), and MHH Hannover (HiLF grant to M.A. and H.K. as well as and support from the young academy to N.L.). M.P.K. and A.S. were funded by the SFB738 "Optimization of conventional and innovative transplants".

## Author contributions

M.A., H.K., R.Z., A.M., and N.L. designed the study, wrote the paper, and performed experiments. C.H., M.H., A.R.H., K.B., J.W.S., K.H., M.P.K., S.G., and C.F. performed experiments and analyzed data. A.S., T.M., S.W., and D.J., K.S., U.M. provided conceptual advice, discussed results, and edited the manuscript.

## Additional information

**Competing interests:** Part of this work is included in a patent application. M.A., H.K., T. M., R.Z., and N.L. are authors of the patent application (European patent application number PCT/EP2018/061574) entitled "Stem-cell derived myeloid cells, generation and use thereof". The priority date of the application is 04.05.2017. All the remaining authors declare no competing interests.

