## [Peer Review File · Nature Communications]

Reviewers' comments:

Reviewer #1 (Remarks to the Author):

The manuscript entitled "Scalable generation of human iPSC derived phagocytes allows for immunotherapy targeting bacterial airway infections" by Ackermann and Kempf et. al. describes a technical approach to upscale the generation of iPSC-derived macrophages for the treatment of respiratory infections. The authors demonstrate a bioreactor driven strategy to increase the expansion of phenotypically, transcriptionally and functionally distinct human macrophages. Generated human macrophages were comprehensively tested in vitro and successfully reduced symptoms of respiratory infections upon transplantation into mice infected with *P. aeruginosa*. Overall, the manuscript is well written, but several issues will need to be addressed:

Major remarks:

- If the administration of iPSC-Mac does help resolve respiratory infections, wouldn't the transplantation of PBMC-Mac lead to the same result? Presented in vivo studies are lacking important controls to draw objective conclusions, determine the clinical potential and evaluate the benefit of iPSC-Mac over regular PBMC-Mac administration.

- o Missing controls/conditions:

- o PBMC-Mac + infection

for functional comparison of iPSC-Mac to PBMC-Mac

are iPSC-Mac as efficient as PBMC-Mac?

Can you reach full clearance with consecutive administration?

- o PBMC-Mac and iPSC-Mac w/o infection

to determine immunological reaction and tolerance of mice for ex vivo generated macrophages

- Does PBMC-Mac or iPSC-Mac application induce unspecific immune responses?

- How long do these cells remain in the lung without infection? Are the cells cleared after a few days?

- Potential side-effects of iPSC-derived cells? Invasion of iPSC-Mac into blood stream? Were other tissues screened for human cells after euthanasia?

- The authors state that PBMC-Mac and iPSC-Mac are "highly similar" regarding their transcriptional changes after pathogen contact (Page 11, Line 14-15). However, the authors also show in Figure 3 that iPSC-derived macrophages demonstrate significantly higher levels of gene expression (Figure 3 c and d), iPSC-Mac highly upregulate a specific gene set (Figure 3e) and biological processes as well as molecular functions for biological functions and responses to cytokines, growth factors and chemokines are specifically lacking in iPSCs. These differences are in contrast to their description of "highly similar".

- o Do these differences result in a different response in vivo compared to PBMC-Mac?

- o Are iPSC-Mac less or more active due to these differences?

Minor remarks:

- Page 5, line 15: Expression of CD14 and CD163 is not "pure" as stated in the text. Cells are obviously very heterogeneous for both markers in Figure S1c

- Page 5, line 17: Legend wrong and reference wrong, should say c-f

- Page 5, line 21: the time line in Figure 1a does not match the text

- Page 7, line 4: lower picture? Text does not match the figure. Should it say right side, or right pictures?

- Page 14, line 16: Reference should say 4c+d?

- Page 15, Figure 4 a: font below mouse is overlapping

- Figure 2 + 3: The two batches of PBMC and iPSC-derived macrophages should be labeled with 1 and 2 to indicate which dataset belongs to which cell product. Without labeling a comparison of data is hard to do.

Reviewer #2 (Remarks to the Author):

This is a well written paper describing the production of iPSC-derived macrophages for use in the treatment of pulmonary infections. Although there are several previous reports describing the production of iPSC-macrophages, this is the first report that I am aware of that uses a stirred tank bioreactor. This allows significant scale up of the procedure and thus more clinically applicable. As iPSC-macrophages are being considered as therapy for a wide range of diseases, this manuscript will be of interest to many fields.

The authors note that iPSC-derived macrophage have a transcriptional profile more similar to primitive yolk sac derived macrophage compared to bone marrow monocyte derived cells. As the authors acknowledge, this has been described by others in murine cells but it is novel that they have found this in the human context too.

The authors use iPSC-macrophage in a mouse model of pulmonary infection and note a significant therapeutic effect with a reduction in lung bacterial numbers and normalisation of rectal temperature and body weight. This has exciting implications for human cell therapy but it is not clear to me now the timing of the mouse experiments compares to the clinical situation. Perhaps this can be clarified in the manuscript.

They do not directly compare iPSC-derived with monocyte derived cells. This is not necessarily a criticism but it might be useful to refer to studies where this has been done in the past (if appropriate)

Specific comments:

Methods

Page 23 line 13: "largest EBs were selected by sedimentation properties"

This is not enough information for others to repeat and more specific details are required. How large and how long are they allow to settle or what centrifugal force? Is this the same strategy that is mentioned on the following page as sedimentation <10mins. It should be clearer exactly what timing this is,

Page 23 line 14: please define "differentiation media" when it is first mentioned here. I assume this is the same as differentiation medium 1 (page 24 line 8). Please define the precise concentrations of IL3 and CSF1 here too.

Results

Macrophages appear smaller and less vacuolated in harvest 1. Is this so – is this because the macrophages phagocytose dead cells in the culture?

Supplementary Figure 1C does not state what harvest this is? Based on the cytopsin, it appears they are harvest no.1? It appears that there are 2 populations of CD11b+ macrophages with a proportion that are CD14-negative and CD163-negative? Has multi-colour flow cytometry analyses been done to assess the heterogeneity of these populations and how that changes with the different harvests? What harvests were used in the in vivo experiments?

Figure 5d. It is interesting that the infected mice are less active compared to the control and infected that are treated with macrophages but it is disappointing that this is only n=1. This should be removed or repeated to generate a significant, reproducible result.

Figure 5f. it appears that there is no significant difference (ns on figure) between infected and infected +PiMT with respect to body weight but the text implies there is a difference (page 16 line 19).

Figure 5i. What do the error bars on the graph represent? The levels of human cytokines have been analysed in 2 mice so statistical analysis cannot be performed. Is this the range? No negative controls are shown and so these values don't actually mean very much as we have nothing to compare to.

Minor corrections/typos

Page 4, line 5: Does 4D refer to the fact that the 3D culture is continually harvested over time? Or does the 4th dimension relate to some other factor? Page 5 line 7 refers to 3D?

Supplementary Figure 1 legend should be (e) should be (f).
Should be consistent as upper or lower case (a) in legend and (A) on actual figure.

Page 16 line 23: authors should be referring to Figure 5h after the sentence " showed reduce erythrocyte levels, compared to infected, non-transplanted controls (Figure 5h). Remove referral to Figure 5h in the next sentence (line 25).

Reviewer #3

This is an impressive technical achievement from Lachmann and colleagues. There are two small points:

1. The authors should consider (in the future) writing a much more in-depth protocol-type manuscript that gives explicit details on how each step of the process. It is clear that the Lachmann group has made far more technical advances in this area than just about any other group. However, the question is whether the technology is accessible to others, or if there is 'voodoo' involved in the methodology.
2. The authors choice of model is intriguing. However, a note should be made that PA is really a problem in chemotherapy where neutrophil numbers decline. Recent work (Kamei et al.) has shown that in neutropenia, alveolar macrophages can afford protection if previously exposed to vaccine strains, and they appear to have characteristics of activated A.M.s. The question for the authors concerns the similarities and differences in the two settings - macrophages are protective, but by what mechanisms. A small adjustment to the discussion section could cover this issue.

We are grateful to the reviewers' comments on our manuscript NCOMMS-18-03992 entitled "*Scalable generation of human iPSC derived phagocytes allows for immunotherapy targeting bacterial airway infection*".

We appreciate the evaluation of our manuscript and are pleased to hear that our approach "*is an impressive technical achievement*" and "*the first report that uses a stirred tank bioreactor*" which "*allows significant scale up the procedure and more clinically applicable*". We moreover appreciate that "*iPSC-macrophages are being considered as therapy for a wide range of disease*" which "*will be of interest to many fields*".

Reviewer's recommendations related to the comparison and classification of iPSC-Mac and PBMC-Mac, in particular regarding cells' *in vivo* functionality; however, R#2 stated that "*this is not necessarily a criticism but it might be useful to refer to studies where this has been done in the past (if appropriate)*". In addition, comments have been directed to the phenotypic characterization of iPSC-Mac and how our mouse model of acute lung infection compares to recently published studies.

Along this line, we would underline the proof-of-concept character of our work, demonstrating the feasibility to continuously produce macrophages especially from human iPSC in stirred tank bioreactors and their therapeutic benefit targeting pulmonary infections.

To further underline these findings, we have slightly modified our manuscript title, which now reads "*Mass production of human iPSC derived macrophages in stirred tank bioreactors enables immunotherapies against bacterial airway infections*"

We have carefully replied to all points raised by reviewers, conducted additional experiments stimulated by the reviewers' comments and thereby comprehensively amended the manuscript. Finally, we corrected typos and introduced minor changes in wording to increase the overall clarity of the text. Please note that in the revised manuscript all changes and specific comments are highlighted in **red**.

Overview of new or revised figures:

Supplementary figure 1

Supplementary figure 3

Supplementary figure 5

Reviewers' comments:

Reviewer #1 (Remarks to the Author):

The manuscript entitled "Scalable generation of human iPSC derived phagocytes allows for immunotherapy targeting bacterial airway infections" by Ackermann and Kempf et. al. describes a technical approach to upscale the generation of iPSC-derived macrophages for the treatment of respiratory infections. The authors demonstrate a bioreactor driven strategy to increase the expansion of phenotypically, transcriptionally and functionally distinct human macrophages. Generated human macrophages were comprehensively tested *in vitro* and successfully reduced symptoms of respiratory infections upon transplantation into mice infected with *P. aeruginosa*. Overall, the manuscript is well written, but several issues will need to be addressed:

Major remarks:

1. If the administration of iPSC-Mac does help resolve respiratory infections, wouldn't the transplantation of PBMC-Mac lead to the same result? Presented *in vivo* studies are lacking important controls to draw objective conclusions, determine the clinical potential and evaluate the benefit of iPSC-Mac over regular PBMC-Mac administration.
2. Missing controls/conditions: PBMC-Mac + infection for functional comparison of iPSC-Mac to PBMC-Mac are iPSC-Mac as efficient as PBMC-Mac?

Response: We thank the reviewer for highlighting these issues. We would like to emphasize that the general intention of our work is to provide efficient means of macrophages mass production using iPSC technology and subsequently enable novel treatment strategies. To the best of our knowledge such techniques are not yet available. Generation of peripheral blood-derived macrophages is currently achieved by the isolation of patient-derived PBMCs or CD14⁺ monocytes and their subsequent differentiation towards macrophages by addition of specific cytokines. As outlined in our discussion, upon clinical translation of our cell therapy approach a 60kg patient would require a cell dose of approx. 1×10^{10} macrophages (extrapolation based on a body weight of 25g per mouse, page 22, line 16-20). Generating this number of macrophages from peripheral blood would require 25-100 liter of fresh blood. Given the limited availability of suitable donors as well as the rather low efficiency and patient-dependent heterogeneity of this approach, the clinical use of such cells is not straightforward.

In contrast, our proposed method paves the way for the continuous and scalable mass production of iPSC-Mac *in vitro* thereby fostering development of novel therapeutic approaches.

Consequently, we have only used PBMC-Mac as a relevant *in vitro* control to benchmark our iPSC-Mac. However, inspired by the reviewer's feedback, we have requested changes to our animal approval and (after allowance) conducted experiments to evaluate the antimicrobial activity of PBMC-Mac *in vivo*. Immunodeficient mice were infected with a low dose of *P. aeruginosa* (PAO1) and subsequently received our experimental cell therapy (pulmonary iPSC-macrophage transplantation; PiMT) by pulmonary administration either of PBMC-Mac or iPSC-Mac (infected only controls received PBS) at 4 hours post infection (non-infected mice served as healthy controls). Notably, as shown in Figure R1a and b, we did not observe a therapeutic effect of PBMC-Mac on the disease score or body temperature. In contrast, our iPSC-Mac demonstrated an improvement of disease-related parameter also in this infection scenario.

Redaction

Redaction

These preliminary results suggest that PBMC-Mac are substantially less potent in clearing bacterial lung infections in our model compared to their iPSC-derived counterparts. Moreover, these observations are in line with our *in vitro* data in Figure 3c and d showing a more pronounced antimicrobial response of iPSC-Mac on the transcriptome level compared to PBMC-Mac.

Notably, however, the focus of the manuscript is to describe the mass production of iPSC-macrophages and their use for novel treatment option. Consequently, and to maintain the focus of our manuscript, we would preferably not include these data in the current manuscript.

3. Can you reach full clearance with consecutive administration?

Response: Addressing full clearance with consecutive administration of macrophages is an intriguing idea. However, we feel that the full treatment effect even of our single-dose cell administration may not be reached within 24 hours. This is indicated by the number of remaining bacteria detectable in the lungs of PiMT treated animals in the therapeutic PiMT scenario ($2.0 \times 10^4 \pm 1.0 \times 10^4$ n=3, mean \pm SD), which also correlates with detectable levels of murine and human pro-inflammatory cytokines (Fig 5i, Supplemental Fig 5a and b) in the bronchio-alveolar lavage fluid (BALF), suggesting a still ongoing antimicrobial response of the immune system. As a consequence, prolonged observation of transplanted animals should be considered first, before evaluating the administration of additional cell doses. To test the idea of prolonged monitoring, we postponed sacrifice of transplanted animals to 72 hours post infection (of note, only infected animals were previously sacrificed 24 hours post infection to prevent extended suffering in line with animal welfare regulations). **Redaction**, the disease score clearly declined at 72 hours compared to the preceding analysis at 24 hours.

Redaction

Redaction

Despite these data, we believe that our xeno-transplant model may not represent an appropriate model to study the repetitive cell-based treatment in sufficient detail. In our model, we applied macrophages to support and replace the endogenous pulmonary immunity. While alveolar macrophages represent the first line of host defence, alveolar macrophages do also orchestrate the adaptive immunity in order to reach full remission. In our study, we provide proof-of-evidence that human iPSC-macrophages can indeed improve the disease symptoms in an immunocompromised, xeno-transplant mouse model of acute pulmonary infections. Therapeutic benefit of our bioreactor-derived macrophages has been demonstrated by either simultaneous or therapeutic administration in mice previously infected with *P. aeruginosa*. Although we could observe a strong therapeutic effect of our cells *in vivo*, we assume that the full magnitude of our therapeutic approach (incl. consecutive administration) could be underestimated in this xeno-transplant model, as our mouse model lacks adaptive immune cells which are normally recruited to the lungs, as a result of infection and the production of proinflammatory cytokines. Taken together, we feel that the employed model will not be suitable to study the necessity/frequency of consecutive administrations.

4. PBMC-Mac and iPSC-Mac w/o infection to determine immunological reaction and tolerance of mice for ex vivo generated macrophages

5. Does PBMC-Mac or iPSC-Mac application induce unspecific immune responses?

Response: We appreciate these comments and agree with the reviewer that immunological reactions and tolerance is an important safety concern for future clinical studies. Similar to the previous question, however, we do not feel comfortable addressing this question within the current xeno-transplantation model. In our study, we have evaluated the therapeutic effect of human cells in an immunocompromised mouse model lacking adaptive immunity. While the genetic background of the applied mouse model represents the gold-standard for testing the therapeutic function of human cells *in vivo*, this model has obvious limitations regarding the role of the adaptive immunity.

Of note, no signs of inflammation after macrophage transplantation were detectable in the same, non-infected mouse model in a previous study by us^{1,2}. However, to properly determine immunological reactions, immunocompetent mice would be mandatory to address the potential side effects of intra-pulmonary delivery of iPSC-macrophages. Since human cells would induce a xenogenic immunological reaction in such mice, murine iPSC-derived macrophages have to be used, to gain relevant knowledge on their immunological properties.

Along these lines, our previous studies using murine iPSC-macrophages in an intra-pulmonary transplantation scheme in diseased *Csf2rb* deficient mice did not reveal any detrimental effects or immunological reactions as a consequence of cell administration³. Follow up for up to four months post transplantation, indicated no sign of immunological reaction. Notably, these experiments used iPSC-Mac from an isogenic mouse strain; thus, further studies employing non-matched murine iPSC-macrophages will be needed to address the topic of allogeneic iPSC-macrophage transplantation to the lung.

Irrespective of these issues, careful evaluation of lung tissue sections has been performed by an experienced pathologist to gain insights into potential side effects of intra-pulmonary human iPSC-macrophages transfer in our current experimental setup. As outlined in the manuscript, massive granulocyte infiltration, severe hemorrhage and alveolar edema have been observed in infected mice only, whereas lung tissue sections of mice receiving iPSC-macrophages were essentially normal. This observation underlines that the procedure of macrophage transplantation is well tolerated and has a high safety profile.

While (un)specific immune response may occur after pulmonary transfer of allogeneic iPSC-macrophages, the use of “universal iPSC lines” or patient-specific iPSC lines may circumvent this problem. In our study, we demonstrate that the mass production of human macrophages in bioreactors is feasible. This technology would ideally collude in concert with genetically engineered universal iPSC lines, which may escape immune rejection and may thus be used as an allogeneic, mass generated “of the shelf” product. Supporting this idea, previous studies have demonstrated the successful generation of iPSC with modified HLA-E surface molecules, supporting the escape from NK-cell mediated clearance.

To further outline the safety of PiMT, we have incorporated a new paragraph in the discussion as follows:

“...This suggests that transplantation of iPSC-Mac is well tolerated and safe, concurring with observations from pulmonary macrophage transplantation studies using macrophages from different stem cell sources¹⁻⁴. Nevertheless, more studies are needed to address potential immunological side effects induced by the transplanted macrophages. Similarly, longer follow-up, detailed dosology as well as intense pre-clinical safety and toxicology studies will be required prior to clinical translation...” (Page 22, line 8-15).

And: *“...To translate this concept to clinical application, either allogeneic or genetically modified universal iPSC⁵ may be employed to produce suitable macrophages in a cost-effective way by industrial scale bioreactor systems as an off-the-shelf product...”* (Page 24, line 5-8).

6. How long do these cells remain in the lung without infection? Are the cells cleared after a few days?

7. Potential side-effects of iPSC-derived cells? Invasion of iPSC-Mac into blood stream? Were other tissues screened for human cells after euthanasia?

Response: In line with the overall biology and the unique functions of tissue resident macrophages, it would be of interest to investigate the persistence of human iPSC-macrophage *in vivo*. Whereas in our acute infection model only a short-term intervention is envisioned, long-term persistence might be important in infection scenarios targeting chronic infections, which are typically observed in cystic fibrosis patients.

In fact, we here provide evidence that macrophages from human iPSC possess a primitive transcriptional fingerprint. Recent studies from our own group and collaboration partners demonstrate that murine and human iPSC or HSC-derived macrophages can adapt an AM phenotype following intra-pulmonary transfer. In these studies, transplanted macrophages could be detected for up to six months post single administration of cells¹⁻⁴. It is worth mentioning that adaptation of transplanted macrophages in these experiments was most likely driven by an open lung tissue microenvironment, which is most likely due to the specific genetic mouse background used. Of note, we have intentionally used a similar mouse model for our experiments, as patient suffering from hereditary alveolar proteinosis are also vulnerable for severe pulmonary infections⁶.

Safety of PiMT is an aspect of utmost relevance. Thus, we would like to point out that there is good evidence for a general high tissue specificity of macrophage lung transplantation in various settings from our several previous studies^{1, 2, 4}. In none of these studies, transplanted cells were detected in organs associated with homing of macrophage, suggesting limited side effects. While we have used the same mouse model, we assume similar cell migration behaviour as previously shown^{1,2}.

However, following reviewers valuable request, we provide additional data employing a sensitive PCR to track the fate of iPSC-derived macrophages post transplantation in a non-diseased scenario. Since the applied hiPSC line was generated by lentiviral reprogramming⁷, we have used specific primer to detect a defined region in the long terminal repeats (LTR) on the lentiviral vector backbone. Employing this method, we analysed different organs of one mouse (d7 post transplantation of 4 Mio iPSC-Mac), which are associated with homing of macrophages. In line with previously published data^{1,3}, we could exclusively detect human cells in the lung and BALF of transplanted animals, whereas no human cells were detectable in Liver, Spleen and BM (Figure R3). Of note, dilutions of human iPSC gDNA were performed in order to define the detection limit of engrafted cells which would be less than 1 in 1000 cells.

Fig R3

Figure R3: PCR-based analysis of tissue-specific engraftment. Purified genomic DNA was used to perform a specific PCR on the 3'LTR of the integrated reprogramming cassette present in the human iPSC. Control primers served as a positive control for the presence of gDNA. Left: Dilution of gDNA from hiPSC cultured on murine feeder cells. Right: Analysis of different organs from animals transplanted with 4 Mio iPSC-Mac and sacrificed after 1 week.

As our data is in line with previous reports¹⁻⁴, we suggest to exclude this data from the current manuscript. Given already published literature, we have incorporated a new section into the discussion in order to provide background information on macrophage cell persistence in the mouse model used and have cited important publications.

The passage now reads "...In our transplantation scenarios, iPSC-Mac were still detected at 24h, however, later time points were not investigated. Earlier studies using the same mouse model demonstrated long-term pulmonary engraftment of cord blood- or iPSC-derived macrophages, while no cells could be detected in other organs associated with homing of macrophages^{1,2}. While the open lung microenvironment may serve as an instructive signal for the local

engraftment of cells, the ancestry of the individual macrophages subset also has to be considered in more detail...” (page 23, line 16-23).

8. The authors state that PBMC-Mac and iPSC-Mac are “highly similar” regarding their transcriptional changes after pathogen contact (Page 11, Line 14-15). However, the authors also show in Figure 3 that iPSC-derived macrophages demonstrate significantly higher levels of gene expression (Figure 3 c and d), iPSC-Mac highly upregulate a specific gene set (Figure 3e) and biological processes as well as molecular functions for biological functions and responses to cytokines, growth factors and chemokines are specifically lacking in iPSCs. These differences are in contrast to their description of “highly similar”.

Response: The reviewer brings up an important topic, highlighting our statement on the iPSC-Mac identity and pointing towards the ‘significantly higher levels of gene expression’ noted in Figure 3c, d. In general, our iPSC-Mac clearly cluster to the PBMC-Mac on a global level, as compared to the clearly divergent expression pattern of undifferentiated iPSC from which they were derived (Figure 2c). However, more detailed analysis revealed distinct gene signatures, including the YS-like identity (Figure 2e, f and g), the increased expression levels of genes associated with the innate immune response (Figure 3d) as well as the more pronounced response following pathogen contact. While these parameters are important attributes and describe the overall transcriptional fingerprint of iPSC-macrophages, clear differences to PBMC-macrophages became apparent.

To describe our data more accurately, we have modified the manuscript as follows:

- Changing ‘*highly similar*’ (Page 12, Line 10) to ‘*similar*’
- In order to emphasize the more pronounced response of the iPSC-Mac we included an additional supplementary Figure (Supplementary Figure 3) illustrating the transcriptional changes of iPSC-Mac versus PBMC-Mac following pathogen contact by principal component analysis (PCA).

9. Do these differences result in a different response *in vivo* compared to PBMC-Mac?

10. Are iPSC-Mac less or more active due to these differences?

Response: We would like to thank the reviewer highlighting these interesting interpretations of our data. However, it may be difficult to provide a clear answer on the underlying mechanism(s) for a macrophage-specific transcriptional signature and the importance of this expression pattern for an effective *in vivo* functionality. Indeed, iPSC-Mac seem to have a similar phagocytic capacity to PBMC-macrophages (Figure 3b), whilst a higher antimicrobial activity was suggested by our transcriptome data (Figure 3c-e). To delineate the superior antimicrobial activity of iPSC-Mac, we have conducted preliminary *in vivo* studies implementing PBMC-Mac transplantation that are seemingly less effective in clearing an acute *P. aeruginosa* infection compared to iPSC-Mac. For a more detailed response, please see also R#1Q#1 and Figure R1 and 2.

Whether the bioreactor-based iPSC-macrophage production (i.e. via shear stress), the primitive YS-like origin of these cells, or even other parameter contribute to the improved antimicrobial activity remains elusive at this point. In fact, analysis of the supernatant from ongoing iPSC-macrophage culture/production media revealed increased levels of pro-inflammatory cytokines such as IL2, IL6, TNF α , and IFN α 2 (Figure 1 e), indicating a pre-activation of the macrophages. Given the complexity of macrophage polarization, activation and antimicrobial function, we feel that there is not sufficient data available yet to draw a clear conclusion. However, our preliminary *in vivo* data (Figure R1 and R2) indeed indicate a higher efficiency of iPSC-Mac in combatting murine *P. aeruginosa* infection.

Minor remarks:

- Page 5, line 15: Expression of CD14 and CD163 is not “pure” as stated in the text. Cells are obviously very heterogeneous for both markers in Figure S1c

Response: We agree with the reviewer’s comment, that the CD14 and CD163 population from the 6-well suspension cultures shown in Supplementary Figure 1c is not pure. However, we would like to emphasize, that the presented flow cytometry data reflect the expression profile of cells that were directly harvested from suspension culture, without conducting terminal differentiation. Most of the CD14- and CD163- cells are thus still immature at this point and will only upregulate expression of respective markers upon terminal differentiation. Strikingly, bioreactor-derived cells

consistently showed higher cell purities with respect to CD14 and CD163 expression even without terminal differentiation, as shown in Figure 2a.

In line with reviewer's suggestions, we rephrased the sentence to '*...exhibited a clear surface marker profile...*' referring to the 6-well suspension cultures and kept '*...a highly pure CD45+CD11b+CD163+CD14+CD34-TRA1-60- phenotype...*' for the bioreactor-derived iPSC-Mac on page 5 line 14-15 and page 8, line 13-15, respectively.

In line with R#2Q#4 we have also included additional data in Supplementary Figure 1.

- Page 5, line 17: Legend wrong and reference wrong, should say c-f

Response: We apologize for the mislabelling of Supplementary Fig 1f. We have amended the manuscript accordingly in the legend and reference.

- Page 5, line 21: the time line in Figure 1a does not match the text

Response: We are grateful for pointing towards this inconsistency. We have now changed the text to '*...From day 10 onwards, weekly harvest of iPSC-Mac from the bioreactors showed an increase in cell yield over time...*' in line with the Figure label.

- Page 7, line 4: lower picture? Text does not match the figure. Should it say right side, or right pictures?

Response: We thank the reviewer for pointing towards the improper Figure legend and changed the text to '*...Images of the 8-blade impeller (right)...*' in line with panel in Figure 1b.

- Page 14, line 16: Reference should say 4c+d?

Response: We thank the reviewer for identifying the missing reference to the Supplementary Figure. We have corrected the reference to Supplementary Fig. 4 accordingly (page 15, line 19).

- Page 15, Figure4 a: font below mouse is overlapping

Response: We thank the reviewer for noting the overlapping label. We have modified the Figure accordingly.

- Figure 2 + 3: The two batches of PBMC and iPSC-derived macrophages should be labeled with 1 and 2 to indicate which dataset belongs to which cell product. Without labeling a comparison of data is hard to do.

Response: We agree that allocation of different data to individual batches is difficult with the previous labelling. We have labelled each sample individually across all heat maps.

Reviewer #2 (Remarks to the Author):

This is a well written paper describing the production of iPSC-derived macrophages for use in the treatment of pulmonary infections. Although there are several previous reports describing the production of iPSC-macrophages, this is the first report that I am aware of that uses a stirred tank bioreactor. This allows significant scale up of the procedure and thus more clinically applicable. As iPSC-macrophages are being considered as therapy for a wide range of diseases, this manuscript will be of interest to many fields.

The authors note that iPSC-derived macrophage have a transcriptional profile more similar to primitive yolk sac derived macrophage compared to bone marrow monocyte derived cells. As the authors acknowledge, this has been described by others in murine cells but it is novel that they have found this in the human context too.

The authors use iPSC-macrophage in a mouse model of pulmonary infection and note a significant therapeutic effect with a reduction in lung bacterial numbers and normalisation of rectal temperature and body weight. This has exciting implications for human cell therapy but it is not clear to me now the timing of the mouse experiments compares to the clinical situation. Perhaps this can be clarified in the manuscript.

Response: We are thankful to reviewers' encouraging comments.

As highlighted in the manuscript and illustrated in Figure 5A, we have intentionally used an acute infection model, reaching a severe and life-threatening disease phenotype already within 24hrs post infection in immunodeficient mice. We would like to underline that the chosen model is a well-established infection model showing lethality within 24-48hrs in immunocompetent mice^{8, 9}. Of note, a similar acute infection model was recently used to evaluate the therapeutic efficacy applying a bacteriophage-based approach to target pulmonary *P. aeruginosa* infections¹⁰. It is noteworthy, that we have accurately dissected different pathogen loads in immunodeficient mice in order to establish the presented acute infection strategy in a humanized mouse model. This acute infection scenario was chosen to (i) show proof of concept for the therapeutic applicability of the concept and (ii) to demonstrate that iPSC-Mac can rescue an acute infection *in vivo*.

The 4hrs post infection time-point for therapeutic application provides an optimal window to evaluate the therapeutic efficiency of our iPSC-Mac. It is noteworthy that the animals received a significant dose of pathogens, which leads to an established infection already after 2-4 hours. Of note, infected but non-transplanted animals (infected control mice) have reached a severe and life-threatening disease score after 24 hours and had to be sacrificed in order to avoid further suffering of mice (according to animal guidelines given by the Federation of European Laboratory Animal Science Associations FELASA).

Direct clinical translation of our approach with respect to a patient-specific cell generation in timing might be challenging. However, given our readily established cell production platform supporting controlled process up-scaling, current efforts are dedicated towards the development of a GMP-compliant cell production process at larger scale. Given the possibility to generate an iPSC-macrophage cell product from genetically enhanced universal iPSC, we envision to establish an off-the-shelf cell product, which can be rigorously tested for safety and subsequently provided for clinical treatments.

In line with reviewer's suggestion, we have added background information on the envisioned clinical transfer of our approach in the discussion as follows: *"To translate this concept to clinical application, either allogeneic or genetically modified universal iPSC⁵ may be employed to produce suitable macrophages in a cost-effective way by industrial scale bioreactor systems as an off-the-shelf product. While our proof-of-concept study is done in an acute infection mouse model, a first application of an iPSC-Mac based therapy (iMATH) approach in humans may be envisioned in CF patients suffering from chronic P. aeruginosa infections. Given the defined clinical background of such patients, first-in-human studies may be performed in a homogenous patient population before this approach is translated to patients suffering from acute and life-threatening respiratory infections..."* (page 24, line 5-14).

1. They do not directly compare iPSC-derived with monocyte derived cells. This is not necessarily a criticism but it might be useful to refer to studies where this has been done in the past (if appropriate)

Response: Reviewer highlights an interesting note. In fact, our presented study represents the first proof-of-concept to derive hematopoietic cells (in particular macrophages) from human iPSC in stirred tank bioreactors and proves their therapeutic use targeting pulmonary infections. As pointed out by R#2, we wanted to provide evidence that iPSC-derivatives can be used as an immunotherapy against *P. aeruginosa*. To the best of our knowledge, our study represents the first cell-based therapy applying macrophages to treat bacterial airway infections, so unfortunately insufficient literature is available on macrophage-based therapies targeting *P. aeruginosa*-mediated infections. From our experience, the use of primary monocyte/macrophages might be challenging with respect to therapeutically required cell numbers and availability (see also R#1 before); we have thus focused on establishing a macrophage farming process using human iPSC in combination with bioreactor technology.

As thoroughly outlined above in response to R#1, we have now conducted *in vivo* experiments comparing iPSC-Mac and PBMC-Mac. Interestingly, our data suggest PBMC-Mac being less effective compared to their iPSC-derived counterparts (Figure R1 and R2), which is in line with our *in vitro* transcriptome data (Figure 3). However, the underlying mechanism for this observation remains elusive. It is well possible that the observed minor differences in the transcriptome and attributed YS-identity of the iPSC-Mac are associated with higher plasticity of the cells and fast adaptation of the macrophages towards the required effector cell against bacterial infection. However, also the process of stirring in the bioreactor-based differentiation process may affect the stage of activation in iPSC-Mac. As evident from Figure 1e, iPSC-Mac already produce a reasonable amount of pro-inflammatory cytokines and thus may represent a pre-activated stage.

A more similar *in vivo* behaviour of human iPSC-derived macrophages to primary CD34⁺-derived macrophages has been previously documented in the context of hereditary pulmonary alveolar proteinosis¹. In this study, the transcriptome of iPSC-macrophages prior and post intra-pulmonary transfer has been compared to macrophages, which have been derived from CD34⁺ cells and transplanted into the same mouse model. Here, transcriptome analysis revealed a similar adaption potential of iPSC-derived macrophages compared to CD34⁺-derived macrophages, highlighting a comparable tissue plasticity.

As suggested by the reviewer, we have incorporated and cited additional background information on the similarities between iPSC-derived macrophages and their CD34⁺ derived counterparts in the discussion part:

"...Previous studies suggested a primitive fingerprint of murine and human iPSC-derived macrophages^{3, 11, 12}, which is further supported by our microarray analysis. Although our cells seem to have a primitive origin, previous studies revealed a similar adaption potential of iPSC-Mac and CD34⁺-Mac, highlighting a similar tissue plasticity¹..." (Page 23, line 23- page 24, line 1).

Specific comments:

Methods

1. Page 23 line 13: "largest EBs were selected by sedimentation properties"

This is not enough information for others to repeat and more specific details are required. How large and how long are they allow to settle or what centrifugal force? Is this the same strategy that is mentioned on the following page as sedimentation <10mins. It should be clearer exactly what timing this is,

Response: We are thankful for highlighting limitations of methods' description. While this technique has not been presented before, we now provide additional information in the material and methods section as follows:

"...After 5 days, EBs were selected by sedimentation properties (sedimentation for 10-15 min) to exclude debris, single cells as well as small cell clusters and subsequently transferred to equilibrated bioreactors..." (page 26, line 13-15).

2. Page 23 line 14: please define "differentiation media" when it is first mentioned here. I assume this is the same as differentiation medium 1 (page 24 line 8). Please define the precise concentrations of IL3 and CSF1 here too.

Response: We apologize for the improper introduction of the differentiation media and missing cytokines concentrations in the manuscript. As described in the previous response (R#2, Q#1) we have completely rephrased the method section of the 'Hematopoietic differentiation in suspension culture' and now provide all required detail enabling trained personnel reproducing the experiments.

Results

3. Macrophages appear smaller and less vacuolated in harvest 1. Is this so – is this because the macrophages phagocytose dead cells in the culture?

Response: We agree with reviewers' notions that macrophages from the first harvest look slightly different on the respective images. However, we do not have clear evidence for changes in quality across different harvests over time. In our differentiation cultures we typically observed relatively big, foamy macrophages with prominent vacuoles. Given their function, we assume that iPSC-Mac - during the generation/differentiation process - may uptake medium or cell debris. Noteworthy, all *in vivo* data were generated from cell harvests No \geq 2. For completeness, we have included this information in the method section as follows on page 34, line 17-18:

'...iPSC-Mac were freshly harvested from suspension cultures after harvest \geq 2...'

4. Supplementary Figure 1C does not state what harvest this is? Based on the cytospin, it appears they are harvest no.1? It appears that there are 2 populations of CD11b⁺ macrophages with a proportion that are CD14-negative and CD163-negative? Has multi-colour flow cytometry analyses been done to assess the heterogeneity of these populations and how that changes with the different harvests? What harvests were used in the *in vivo* experiments?

Response: As stated in the responses to R#1 (minor remarks), it is important to notice that the flow cytometry analysis of macrophages was performed directly from suspension cultures without terminal differentiation. As a consequence, some of the cells remain rather immature and are partially lacking CD14 and CD163, which is not observed after terminal differentiation (new data in Supplementary Figure 1e, see comment below). The flow cytometry data thus indicate a heterogeneous mix of immature myeloid cells and macrophages of relatively high purity.

Noteworthy, this opposes findings from bioreactor harvests, where the same panel of markers showed higher homogeneity of freshly harvested cells. This hints towards an acceleration of maturation of the bioreactor approach, potentially due to the hydrodynamic conditions in impeller-stirred vessels. As indicated in the previous section, we used freshly harvested macrophages from harvest No \geq 2 onwards for *in vivo* experiments and specified this in the method section in the amended manuscript.

With respect to the multi-colour flow cytometry of suspension-based differentiation cultures, we are able to identify two distinct cell populations: (i) CD45⁺/CD11b⁺/CD14⁻/CD163⁻ more immature myeloid cells and (ii) CD45⁺/CD11b⁺/CD14⁺/CD163⁺ macrophages. These immature myeloid cells could be further differentiated towards a homogenous population of CD45⁺/CD11b⁺/CD14⁺/CD163⁺ macrophages in the presence of 50ng/ml M-CSF for seven more days.

The new data has been added to the manuscript in Supplementary Figure 1.

The corresponding paragraph now reads: *"...Generated iPSC-Mac exhibited a clear surface marker profile of CD45⁺CD11b⁺CD14⁺CD163⁺CD34⁻TRA1-60⁻. Of note, freshly harvested cells comprised a minor population of CD45⁺/CD11b⁺/CD14⁻/CD163⁻ immature myeloid cells (Supplementary Fig. 1c and d). Following terminal differentiation for seven more days, iPSC-Mac represented a homogenous population of CD45⁺CD11b⁺CD14⁺CD163⁺CD34⁻TRA1-60⁻ cells with classical macrophage-like morphology and efficiently phagocytosed fluorescently labelled E. coli particles (Supplementary Fig. 1e-g)...."* (page 5, line 14-21)

5. Figure 5d. It is interesting that the infected mice are less active compared to the control and infected that are treated with macrophages but it is disappointing that this is only n=1. This should be removed or repeated to generate a significant, reproducible result.

Response: We were indeed surprised to see that treated animals showed the same motility as control mice. This observation holds true for all mice receiving the therapeutic PiMT. However, we agree with the reviewer that

quantification of mouse motility is not sufficient for $n=1$. Unfortunately, we are lacking equivalent, movie-based data to perform this analysis for additional animals. Consequently, we have removed the quantitative data from the manuscript. However, as the observation was striking, we feel that it is valuable to communicate the trajectory analysis in the paper, if this meets reviewers and editors' agreement.

6. Figure 5f. it appears that there is no significant difference (ns on figure) between infected and infected +PiMT with respect to body weight but the text implies there is a difference (page 16 line 19).

Response: We agree with the reviewer's comment that the phrasing on the body weight is misleading as no statistical difference between infected/infected+PiMT or infected+PiMT/control is indicated. However, we would like to highlight that there is a clear trend towards a beneficial effect on the body weight in PiMT treated animals compared to mice infected only. This observation goes in line with improved values for body temperature after treatment.

To avoid misleading statements, we have re-phrased the text which now reads as follows:

"...Efficiency of therapeutic PiMT was further documented by normalized rectal temperature and a tendency towards maintained body weight values 24h post-infection (Fig. 5e and 5f). Of note, a profound reduction in lung bacterial numbers in infected+PiMT mice was observed (Fig. 5g)..." on page 17, line 19-22.

7. Figure 5i. What do the error bars on the graph represent? The levels of human cytokines have been analysed in 2 mice so statistical analysis cannot be performed. Is this the range? No negative controls are shown and so these values don't actually mean very much as we have nothing to compare to.

Response: We apologize for the imprecise description of Figure 5i. Indeed, the Figure represents the cytokine levels in the BALF of 2 animals receiving PiMT. We performed technical duplicates of this measurements and present these 4 values as mean +/- SD. In our experiments, we performed the cytokine analysis in not-infected control, infected, and infected+PiMT animals ($n=2$ animals each group + technical duplicates). While the human cytokines (hIL-1b, hIL-6, hIL-8, hIFNa2, hMCP-1, and hTNFa) could hardly be detected in non-transplanted animals, we initially decided to present only data from mice which received a therapeutic cell dose.

However, we agree with the reviewer and present a new supplementary Figure 5, demonstrating the respective human cytokine values in not-infected control, infected, and infected+PiMT animals. Moreover, we also performed the analysis of the murine counterparts (IL1b, IL6, MCP1 and TNFa) and present the respective concentration of murine cytokines in the same supplementary Figure.

To highlight the new data, we have modified the corresponding passage and demonstrate the data in Supplementary Figure 5.

The text now reads: *"...This observation was accompanied by the detection of important pro-inflammatory human cytokines such as hIL6, hIL8, hINFa2, hMCP-1 and TNFa only in animals receiving a PiMT, whereas human cytokines could not be detected in control or infected only animals (Fig. 5i and Supplementary Fig. 5a). In contrast, levels for murine pro-inflammatory cytokines (mIL1b, mIL6, mMCP-1, mTNFa) were highly elevated in infected only animals, whereas almost normal levels could be detected in PiMT-treated animals (Supplementary Fig. 5b)..."*. On page 17 line 25 to page 18, line 5

Moreover, in line with the Nature Communication guidelines, we have changed all graphs representing $n<5$ and present individual data points.

Minor corrections/typos

Page 4, line 5: Does 4D refer to the fact that the 3D culture is continually harvested over time? Or does the 4th dimension relate to some other factor? Page 5 line 7 refers to 3D?

Response: We agree with reviewer's notion that the 4-dimensions of the protocol were not properly described and inconsistent in the manuscript. Indeed, the 4th dimension refers to the continuous production of macrophages over time.

We now introduced a consistent 4D labelling specified as follows:

Page 5 line 7-8: '*...we developed a suspension-based (3D), continuous (4D) hematopoietic differentiation protocol, ...*'

Supplementary Figure 1 legend should be (e) should be (f).

Response: We apologize for the mislabelling and modified the manuscript accordingly.

Should be consistent as upper or lower case (a) in legend and (A) on actual figure.

Response: We thank the review for this note and adjusted the labelling of all supplementary Figures to lower case characters.

Page 16 line 23: authors should be referring to Figure 5h after the sentence "..... showed reduce erythrocyte levels, compared to infected, non-transplanted controls (Figure 5h). Remove referral to Figure 5h in the next sentence (line 25).

Response: We thank the review for this note and modified the manuscript accordingly.

Reviewer #3 (Remarks to Authors)

This is an impressive technical achievement from Lachmann and colleagues. There are two small points:

1. The authors should consider (in the future) writing a much more in-depth protocol-type manuscript that gives explicit details on how each step of the process. It is clear that the Lachmann group has made far more technical advances in this area than just about any other group. However, the question is whether the technology is accessible to others, or if there is 'voodoo' involved in the methodology.

Response: We are thankful for the reviewer's comment and highly appreciate the reviewer's view on our overall work. We will ensure that the technology for continuous macrophage production at large scale is accessible to the broad readership and scientific community by providing a detailed step-by-step protocol in the near future. In fact, the laboratory already welcomes collaborators, which would like to learn this technique. This being said, we have very solid indication that the basic protocol is highly robust and reproducible to be established in independent laboratories. Importantly, given the experimental description of the current manuscript, a lab with basic knowledge in hiPSC culture and differentiation should be able to set up the protocol. Towards this end, we have added considerable detail in the method section in response to reviewer comments, particularly regarding the setup of the suspension differentiation.

While we have focused on the mass production of macrophages in this paper draft, we have robust evidence that the technology is also suitable for the mass generation of other cells from the myeloid lineage (own unpublished data). Given the broad applicability, this technique may be of interest to the various disciplines. While the laboratory has made profound progress in the development of a GMP-compliant, bioreactor mediated mass production technology, we request a follow up manuscript with a more detailed step-by-step description. This platform would allow us to provide sufficient details of the method and to describe critical bottlenecks, which are important to successfully establish this innovative method.

2. The authors choice of model is intriguing. However, a note should be made that PA is really a problem in chemotherapy where neutrophil numbers decline.. Recent work (Kamei et al.) has shown that in neutropenia, alveolar macrophages can afford protection if previously exposed to vaccine strains, and they appear to have characteristics of activated A.M.s. The question for the authors concerns the similarities and differences in the two settings - macrophages are protective, but by what mechanisms. A small adjustment to the discussion section could cover this issue.

Response: We are thankful for the reviewers inspiring comment on fighting *P. aeruginosa* infection associated with chemotherapy-induced neutropenia using "vaccine-induced macrophages". In fact, the study of Kamei and colleagues hypothesized the induction of vaccine-induced macrophages (ViMs) by remodelling the endogenous tissue resident macrophage cell pool. Given our iPSC-Mac produced by the bioreactor, it would be an interesting approach to remodel iPSC-derived macrophages towards a ViMs phenotype and to investigate the adoptive transfer and the protective effect of these cells *in vivo*.

In our model, we hypothesize that the therapeutic effect results from increased numbers of macrophages transplanted. In fact, the murine lung microenvironment harbours approx. 3-500.000 AMs/mouse lung¹³. While numbers of innate immune cells steadily increase upon pulmonary infection, we inject 4-8 fold of macrophages in one transplantation dose. Given the high phagocytic capacity (Fig 3b) of our cells as well as the lack of adaptive immunity in our mouse model, we hypothesis a direct effect of iPSC-Mac post transplantation.

The afore mentioned rational has been added to the discussion:

"...As previously shown for endogenous alveolar macrophages¹⁴, iPSC-Mac might be induced towards a vaccine-induced macrophage (ViM) phenotype, which upon intra-pulmonary administration could protect mice from P. aeruginosa infections. This might be of particular interest in life-threatening, chemotherapy associated infections. While in our "protective" regimen, we have used the simultaneous administration of iPSC-Mac and P. aeruginosa, administration of iPSC-derived ViM prior the infection would provide additional insights in the protective properties..." (Page 23, Line 7-14).

References:

1. Happle, C. *et al.* Pulmonary Transplantation of Human Induced Pluripotent Stem Cell-derived Macrophages Ameliorates Pulmonary Alveolar Proteinosis. *American journal of respiratory and critical care medicine* **198**, 350-360 (2018).
2. Happle, C. *et al.* Pulmonary transplantation of macrophage progenitors as effective and long-lasting therapy for hereditary pulmonary alveolar proteinosis. *Science translational medicine* **6**, 250ra113 (2014).
3. Mucci, A. *et al.* iPSC-Derived Macrophages Effectively Treat Pulmonary Alveolar Proteinosis in Csf2rb-Deficient Mice. *Stem cell reports* (2018).
4. Suzuki, T. *et al.* Pulmonary macrophage transplantation therapy. *Nature* **514**, 450-454 (2014).
5. Gornalusse, G.G. *et al.* HLA-E-expressing pluripotent stem cells escape allogeneic responses and lysis by NK cells. *Nature biotechnology* **35**, 765-772 (2017).
6. Suzuki, T. *et al.* Familial pulmonary alveolar proteinosis caused by mutations in CSF2RA. *The Journal of experimental medicine* **205**, 2703-2710 (2008).
7. Lachmann, N. *et al.* Gene correction of human induced pluripotent stem cells repairs the cellular phenotype in pulmonary alveolar proteinosis. *American journal of respiratory and critical care medicine* **189**, 167-182 (2014).
8. Klockgether, J. *et al.* Genome diversity of Pseudomonas aeruginosa PAO1 laboratory strains. *J Bacteriol* **192**, 1113-1121 (2010).
9. Munder, A. & Tummeler, B. Assessing Pseudomonas virulence using mammalian models: acute infection model. *Methods in molecular biology* **1149**, 773-791 (2014).
10. Agarwal, R. *et al.* Inhaled bacteriophage-loaded polymeric microparticles ameliorate acute lung infections. *Nature Biomedical Engineering* (2018).
11. Buchrieser, J., James, W. & Moore, M.D. Human Induced Pluripotent Stem Cell-Derived Macrophages Share Ontogeny with MYB-Independent Tissue-Resident Macrophages. *Stem cell reports* **8**, 334-345 (2017).
12. Takata, K. *et al.* Induced-Pluripotent-Stem-Cell-Derived Primitive Macrophages Provide a Platform for Modeling Tissue-Resident Macrophage Differentiation and Function. *Immunity* **47**, 183-198 e186 (2017).
13. Zhang, X., Goncalves, R. & Mosser, D.M. The isolation and characterization of murine macrophages. *Curr Protoc Immunol* **Chapter 14**, Unit 14 11 (2008).
14. Kamei, A. *et al.* Exogenous remodeling of lung resident macrophages protects against infectious consequences of bone marrow-suppressive chemotherapy. *Proceedings of the National Academy of Sciences of the United States of America* **113**, E6153-E6161 (2016).

REVIEWERS' COMMENTS:

Reviewer #1 (Remarks to the Author):

The authors adequately addressed all of my comments / critiques. Figures have been corrected/changed as recommended, additional data has been implemented as requested, many open questions were appropriately addressed in the discussion, and the methods section is significantly extended.

Reviewer #2 (Remarks to the Author):

All my comments have been addressed in full by the authors

Reviewer #3 (Remarks to the Author):

The authors have done a comprehensive and respectful effort to revise their manuscript, which is now greatly improved.